# The Rad51 paralogs facilitate a novel DNA strand specific damage tolerance pathway

Joel C. Rosenbaum [1,11], Braulio Bonilla[2,11], Sarah R. Hengel[2,11], Tony M. Mertz[3], Benjamin W. Herken[2], Hinke G. Kazemier[4], Catherine A. Pressimone[2], Timothy C. Ratterman [5], Ellen MacNary[3], Alessio De Magis[6], Youngho Kwon[7,8], Stephen K. Godin[2], Bennett Van Houten [9], Daniel P. Normolle[10], Patrick Sung[7,8], Subha R. Das [5], Katrin Paeschke [4,6], Steven A. Roberts[3], Andrew P. VanDemark [1] & Kara A. Bernstein [2]

Accurate DNA replication is essential for genomic stability and cancer prevention. Homologous recombination is important for high-fidelity DNA damage tolerance during replication. How the homologous recombination machinery is recruited to replication intermediates is unknown. Here, we provide evidence that a Rad51 paralog-containing complex, the budding yeast Shu complex, directly recognizes and enables tolerance of predominantly lagging strand abasic sites. We show that the Shu complex becomes chromatin associated when cells accumulate abasic sites during S phase. We also demonstrate that purified recombinant Shu complex recognizes an abasic analog on a double-flap substrate, which prevents AP endonuclease activity and endonuclease-induced double-strand break formation. Shu complex DNA binding mutants are sensitive to methyl methanesulfonate, are not chromatin enriched, and exhibit increased mutation rates. We propose a role for the Shu complex in recognizing abasic sites at replication intermediates, where it recruits the homologous recombination machinery to mediate strand specific damage tolerance.

[1] University of Pittsburgh, Department of Biological Sciences Pittsburgh, Pittsburgh, PA 15260, USA. [2] University of Pittsburgh, School of Medicine, Department of Microbiology and Molecular Genetics, Pittsburgh, PA 15213, USA. [3] Washington State University, School of Molecular Biosciences and Center for Reproductive Biology, College of Veterinary Medicine, Pullman, WA 99164, USA. [4] University of Groningen, University Medical Center Groningen, European Research Institute for the Biology of Ageing, 9713 AV Groningen, Netherlands. [5] Carnegie Mellon University, Department of Chemistry and Center for Nucleic Acids Science & Technology, Pittsburgh, PA 15213, USA. [6] Department of Oncology, Hematology and Rheumatology, University Hospital Bonn, Bonn, Germany. [7] Yale University, School of Medicine, Department of Molecular Biophysics and Biochemistry, New Haven, CT 06511, USA. [8] University of Texas Health Science Center at San Antonio, Department of Biochemistry and Structural Biology, San Antonio, TX 78229, USA. [9] University of Pittsburgh, School of Medicine, Department of Pharmacology and Chemical Biology, Pittsburgh, PA 15213, USA. [10] University of Pittsburgh, School of Public Health, Department of Biostatistics, Pittsburgh, PA 15261, USA. [11] These authors contributed equally: Joel C. Rosenbaum, Braulio Bonilla, Sarah R. Hengel. Correspondence and requests for materials should be addressed to K.A.B. (email: karab@pitt.edu)

DNA is constantly damaged by endogenous and exogenous sources such as alkylating agents, reactive oxygen species, and radiation. Each type of DNA damage is recognized and repaired using a specialized repair pathway. Repair of DNA base damage by base excision repair (BER) begins with recognition and excision of the damaged base by a DNA glycosylase resulting in abasic [also known as apurinic/apyrimidinic (AP)] site formation. In mammalian cells, spontaneous depurination events and repair of endogenous DNA damage generates between 10,000 and 30,000 abasic sites per day[1–3], making them one of the most common genotoxic lesions. Most abasic sites are repaired in an high-fidelity manner by the subsequent steps of BER. During replication, abasic sites are strong blocks to the replication DNA polymerases epsilon and delta[4,5]. When synthesis at a replication fork is blocked by an abasic site, the lesion must be bypassed. Abasic sites within the context of DNA replication are often resolved by either low-fidelity translesion DNA synthesis (TLS)[5] or high-fidelity homologous recombination (HR)[6]. How abasic sites at stalled replication forks are targeted to distinct bypass/ repair pathways remains largely unknown.

The Rad51 paralogs are a highly conserved family of proteins structurally similar to the central HR protein, Rad51 (ref. [7]). The Rad51 paralogs form sub-complexes that aid in Rad51 filament formation and strand invasion, two key steps in HR. Mutations in the human RAD51 paralogs are associated with predisposition to breast and ovarian cancer as well as Fanconi anemia-like syndromes[8,9]. The Shu complex is an evolutionarily conserved complex, which contains Rad51 paralogs. The Saccharomyces cerevisiae Shu complex is a heterotetramer composed of Shu2 (a SWIM-domain-containing protein) and the Rad51 paralogs Csm2, Psy3, and Shu1 (refs. [10–13]). Shu complex mutant cells are especially sensitive to the alkylating agent, methyl methanesulfonate (MMS), which among other agents causes replication blocking lesions, suggesting that the Shu complex may help facilitate their repair[10,14–17].

Here we provide evidence that DNA-binding components of the budding yeast Shu complex, Csm2-Psy3, bind double-flap DNA substrates containing an abasic site analog, and increase chromatin association when abasic sites accumulate. Importantly, Csm2-Psy3 blocks AP endonuclease cleavage at a double-flap DNA substrate, thus preventing in vitro DSB formation. Furthermore, we show that Csm2-Psy3 also aids in preventing TLS-induced mutations that arise in the lagging strand during replication of DNA templates containing abasic sites. Therefore, we propose a model whereby the Shu complex directly recognizes abasic sites on the lagging strand of a replication fork to facilitate a error-free, strand-specific damage tolerance pathway.

## Results

### Csm2 is the primary DNA-binding subunit.
To determine the role of DNA binding in S. cerevisiae Shu complex function, we modeled the putative DNA-binding loops of the Shu complex members, the Rad51 paralogs, Csm2 and Psy3 (ref. [18]). We mutated the lysine and arginine residues within these predicted DNA-binding loops (for Csm2: K189A, R190A, R191A, R192A; csm2-KRRR) (for Psy3: K199A, R200A, K201A; psy3-KRK)[12] (Fig. 1a). To assess the DNA-binding capabilities of Csm2-KRRR and Psy3-KRK, we co-expressed and purified Csm2-Psy3, Csm2-Psy3-KRK, Csm2-KRRR-Psy3, and Csm2-KRRR-Psy3-KRK complexes from Escherichia coli and assessed their capacity to bind their preferred DNA substrate, double-flap DNA substrate or Y DNA[19], by fluorescence polarization anisotropy (Fig. 1b, Supplementary Fig. 1, Supplementary Table 1). Whereas Csm2-Psy3 binds the double-flap DNA substrate with an equilibrium dissociation constant ($K_d$) of $435 \pm 37$ nM, Csm2-KRRR-Psy3,

and Csm2-KRRR-Psy3-KRK mutant proteins exhibit minimal DNA binding, while Csm2-Psy3-KRK exhibits more than a six-fold reduction in DNA-binding affinity relative to Csm2-Psy3 ($K_d > 2.8 \mu$M; Fig. 1b, Supplementary Table 1). These results indicate that Csm2 is the primary DNA-binding subunit.

### Csm2-Psy3 DNA binding is necessary for repair in vivo.
We next asked whether the Csm2-Psy3 DNA-binding activity would be important for their function in vivo. To address this question, we analyzed S. cerevisiae cells expressing csm2-KRRR and psy3-KRK DNA-binding mutants for MMS sensitivity. We observe very modest MMS sensitivity of csm2-KRRR cells, while psy3-KRK cells are largely insensitive to 0.02% MMS, and the csm2-KRRR psy3-KRK double-mutant cells exhibit increased MMS sensitivity and reduced viability compared to the single mutants (Fig. 1c, Supplementary Fig. 2). We next examined whether Csm2 and Psy3 DNA binding would be important for suppressing mutations by measuring CAN1 mutation rates (Fig. 1d). Similar to the MMS sensitivity, we find that csm2-KRRR psy3-KRK double mutant cells exhibit increased spontaneous or MMS-induced mutation rates compared to wild-type cells (Fig. 1d). We next used western blot analysis to ensure that the phenotypes we observed in csm2-KRRR and/or psy3-KRK cells are not due to altered protein expression (Supplementary Fig. 3a). Similarly, we do not observe changes in Shu complex integrity or known protein interactions by yeast-2-hybrid or during recombinant protein purification, where the Csm2-KRRR Psy3-KRK elution profile is similar to wild-type complexes (Supplementary Fig. 3b–d). Therefore, Csm2 and Psy3 DNA binding residues are important for Shu complex function without affecting complex formation. Furthermore, our findings suggest that the combined DNA-binding activities of Csm2 and Psy3 are critical for MMS resistance and suppressing mutations.

### Csm2 is chromatin enriched when abasic sites accumulate.
We find that disruption of the DNA-binding activities of Csm2 and Psy3 leads to increased mutation rates (Fig. 1d) and our previous work shows that when abasic sites accumulate in csm2Δ cells, mutation rates increase over 1000-fold[14]. Therefore, we wanted to determine if Csm2-Psy3 DNA binding is critical for MMS resistance in vivo when abasic sites accumulate. Abasic sites can be forced to accumulate by deleting the enzymes responsible for their processing, which include the AP endonucleases (APN1 APN2) and AP lyases (NTG1 NTG2). Suggesting that Csm2-Psy3 DNA-binding activities are important when abasic sites accumulate, we observe that csm2-KRRR psy3-KRK apn1Δ apn2Δ ntg1Δ ntg2Δ cells exhibit increased MMS sensitivity that is comparable to a csm2Δ apn1Δ apn2Δ ntg1Δ ntg2Δ cell (Fig. 2a).

Since we find that the Shu complex binds most tightly to double-flap DNA[12,19] and is important for resistance to abasic sites (Fig. 2a), we hypothesized that Csm2-Psy3 may be enriched at chromatin when abasic sites accumulate during replication. To test this hypothesis, we performed chromatin fractionation experiments. We first arrested Csm2-6HA-expressing cells (with or without apn1Δ apn2Δ ntg1Δ ntg2Δ) in G1 with alpha factor and released these cells into 0.02% MMS for 1 h before lysis and fractionation. We observe that Csm2 chromatin association increases 4.5-fold when abasic sites accumulate and this enrichment depends on Csm2 DNA-binding activity (Fig. 2b). In contrast to Csm2, we find that RPA chromatin association occurs independently of Csm2 DNA-binding activity (Supplementary Fig. 4a).

We next examined whether Csm2 chromatin association increases in an MMS dose-dependent manner. To test this, we treated Csm2-6HA-expressing cells (with or without apn1Δ

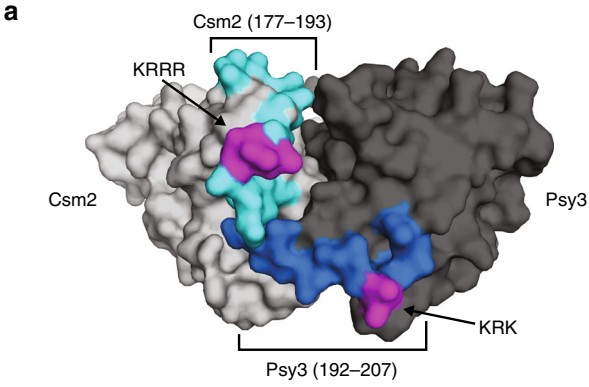

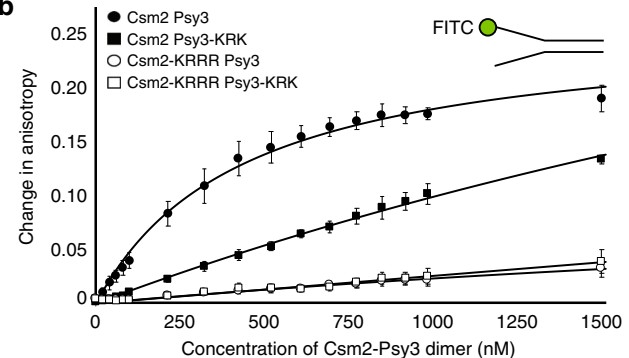

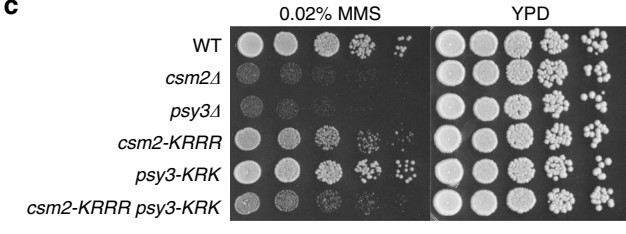

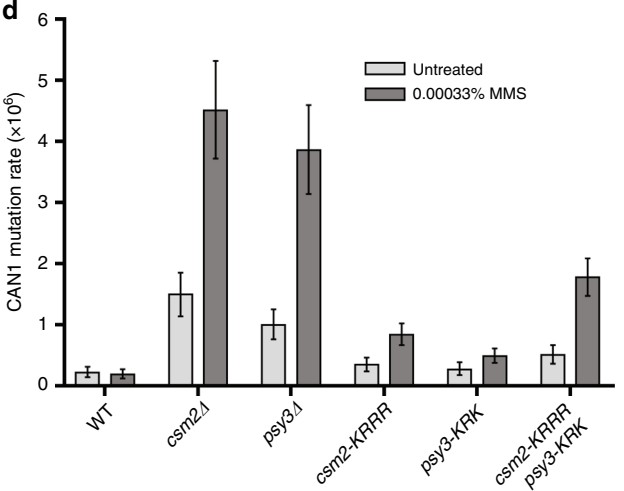

**Fig. 1** Csm2-Psy3 DNA binding is important for Shu complex function. **a** Surface view of *S. cerevisiae* Csm2 (light gray; K189, R190, R191, R192) and Psy3 (dark gray; K199, R200, K201) with the predicted DNA-binding residues highlighted in magenta and predicted DNA-binding loops in light and dark blue, respectively (ref. [12]; model structure derived from PDB 3VU9). **b** In vitro analysis of Csm2-Psy3 binding to a DNA fork substrate compared to Csm2-Psy3 DNA-binding mutants (Csm2-K189A/R190A/R191A/R192A and/or Psy3-K199A/R200A/K201A) by fluorescence anisotropy. Increasing concentrations of Csm2-Psy3 or the indicated mutants were added to 25 nM 3'-fluorescein-labeled double-flap substrate and DNA binding was assessed. Dissociation constants ($K_d$) and associated standard deviations from triplicate experiments were determined by non-linear curve fitting to a one-site binding model. **c** Cells expressing the *csm2-KRRR psy3-KRK* double mutant exhibit increased MMS sensitivity relative to *csm2-KRRR* or *psy3-KRK* cells. The DNA-binding residues shown in **a** were mutated to alanines and integrated into the genomic *CSM2* and *PSY3* loci. Fivefold serial dilution of WT, *csm2Δ*, *psy3Δ*, *csm2-KRRR*, *psy3-KRK*, and *csm2-KRRR psy3-KRK* cells onto rich YPD medium or YPD medium containing 0.02% MMS were incubated for 2 days at 30 °C prior to being photographed. **d** Spontaneous and MMS-induced mutation rate at the *CAN1* locus were measured in WT, *csm2Δ*, *psy3Δ*, *csm2-KRRR*, *psy3-KRK*, and *csm2-KRRR psy3-KRK* cells. Error bars indicate 95% confidence intervals

specificity for MMS-induced damage and abasic sites, we do not observe Csm2 enrichment when forks are stalled with HU (Supplementary Fig. 4b). Overall, these results suggest that the Shu complex is enriched at chromatin when abasic sites accumulate.

**Csm2-Psy3 suppress lagging strand abasic site mutations.** To determine if Csm2 and Psy3 facilitate the bypass of abasic sites, we assessed how disruption of these genes influence the *CAN1* mutation rate and spectrum induced by the human cytidine deaminase, APOBEC3B. APOBEC family cytidine deaminases induce genomic hypermutation in human tumors[20–22]. Bioinformatic analysis of mutations in cancer genomes[23–25] and experiments in yeast[26] and bacterial systems[27] indicate that APOBECs deaminate the lagging strand template during DNA replication. APOBEC3B-induced mutation rates and spectra were previously measured within yeast with *CAN1* at a location 16 kB centromere proximal to ARS216. In this setting, APOBEC-induced mutations occur primarily at G bases in leftward moving forks due to deamination of cytidines on the lagging DNA strand[26,28] (Fig. 3a). Moreover, the APOBEC3B-induced dU is efficiently removed by the uracil glycosylase, Ung1, which results in formation of synthesis-blocking abasic sites on the template of Okazaki fragments (Fig. 3a). We used this system to determine if Csm2 and Psy3 facilitate bypass of abasic sites induced by APOBEC3B during replication of the lagging strand in vivo. We find that combining APOBEC3B expression with Shu complex defects results in a synergistic increase in *CAN1* mutation rates to levels observed in *ung1* deletion strains, in which all APOBEC3B-induced lesions are converted to mutations (Fig. 3b). Importantly, *CSM2* or *PSY3* deletion in combination with *ung1Δ* results in mutation rates similar to the *ung1* single deletion (Fig. 3b), indicating that Shu complex genes are epistatic with *UNG1* in their ability to decrease APOBEC3B-induced mutation. Sequencing of *can1* mutants from *csm2Δ ung1Δ* produced nearly exclusively G to A transitions (Fig. 3c), confirming that Ung1 operates prior to Csm2 in avoiding APOBEC3B-induced mutations. In contrast, *CAN1* mutation spectra in both *csm2Δ* and *psy3Δ* cells revealed both G to C transversions and G to A transitions, consistent with mutations caused by Rev1-mediated and A-rule polymerase-mediated TLS past abasic sites[28]

*apn2Δ ntg1Δ ntg2Δ*) with different MMS concentrations for 1 h (0%, 0.01%, 0.02%, and 0.03%). Importantly, we observe Csm2 chromatin association increases in an MMS dose-dependent manner (~2- to 7-fold) when abasic sites accumulate (Fig. 2c; *p* = 0.02 for 0.02% MMS and *p* = 0.01 for 0.03% MMS). We also observe a reproducible, although not statistically significant, two-fold increase in Csm2 chromatin association in WT cells comparing untreated to 0.03% MMS (Fig. 2c). Consistent with

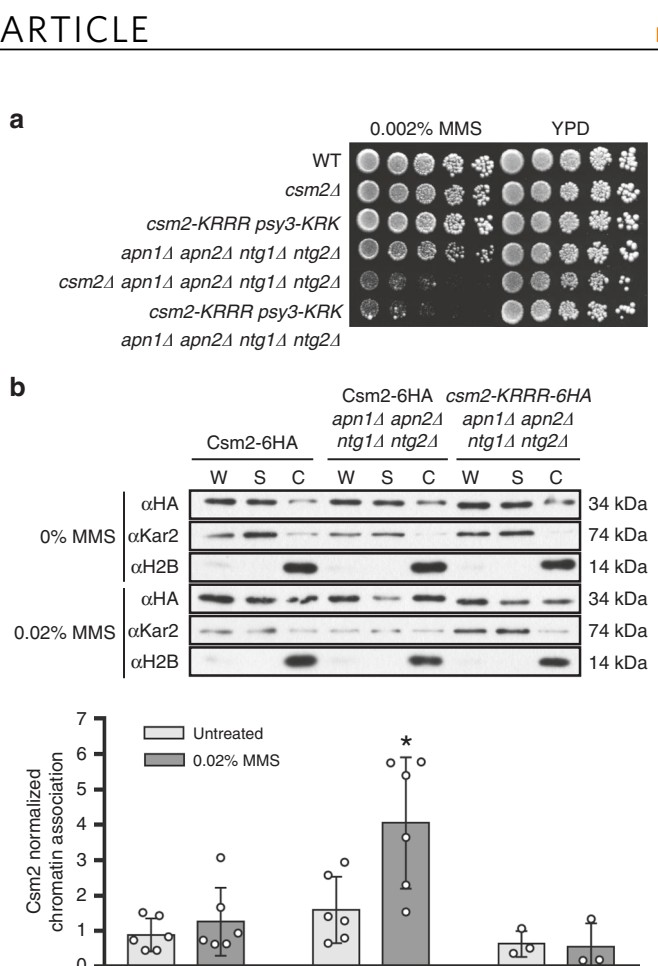

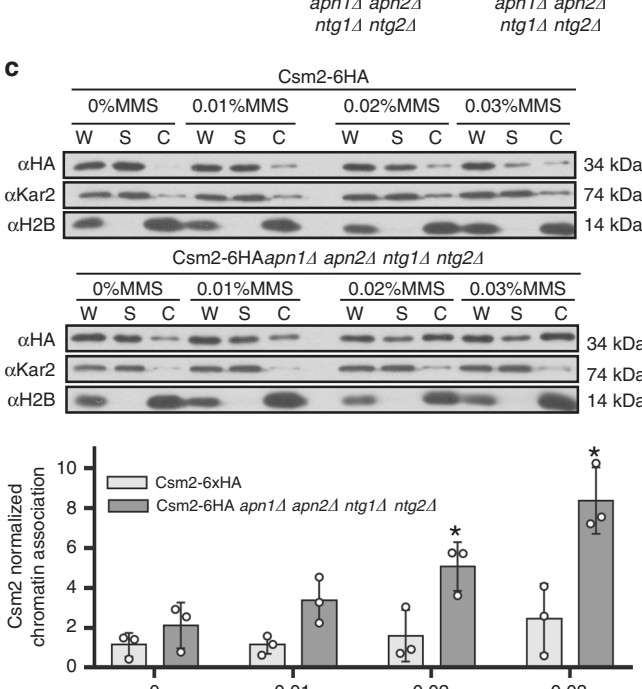

**Fig. 2** Csm2 is recruited to chromatin when abasic sites accumulate. **a** Csm2-Psy3 DNA binding is critical for survival when abasic sites accumulate. Fivefold serial dilutions of the indicated yeast strains on rich medium (YPD) or rich medium containing 0.002% MMS. Abasic sites accumulate by combined disruption of the AP endonucleases (*APN1, APN2*) and AP lyases (*NTG1, NTG2*) in the presence of MMS. Csm2-Psy3 double DNA-binding mutant (*csm2-KRRR psy3-KRK*) exhibits similar MMS sensitivity to *csm2Δ* cells when abasic sites accumulate. **b** Csm2 is enriched at the chromatin when abasic sites accumulate in a DNA-binding-dependent manner. Csm2-6HA-expressing cells were synchronized in G1 with alpha factor and released into YPD medium or YPD medium containing 0.02% MMS for 1 h before cellular fractionation. Csm2 protein levels from whole-cell extract (W), supernatant (S), and chromatin (C) fractions from the indicated strains were determined by western blot using HA antibody. Kar2 and histone H2B were used as fractionation controls (S and C, respectively). The results from three to five experiments were plotted with standard deviations, as fold enrichment relative to the untreated WT (Csm2-6HA). The *p* value between Csm2-6HA *apn1Δ apn2Δ ntg1Δ ntg2Δ* and WT (treated and untreated) or csm2-KRRR-6HA *apn1Δ apn2Δ ntg1Δ ntg2Δ* was calculated using an unpaired two-tailed Student's *t*-test between experimental samples and in each case was *p* ≤ 0.05. **c** Csm2 chromatin association increases in an MMS dose-dependent manner. Same as **b** except that Csm2-6HA or Csm2-6HA *apn1Δ apn2Δ ntg1Δ ntg2Δ* were treated with 0%, 0.01%, 0.02%, or 0.03% MMS and results were quantified as described in **b**

Shu complex promotes an error-free template switch mechanism to inhibit the conversion of abasic sites in the lagging strand template to mutations.

**Csm2-Psy3 binds to abasic site analogs in double-flap DNA.** Since we observe that the Shu complex is important for error-free bypass of abasic sites during replication, we asked if Csm2-Psy3 could directly recognize abasic sites in a double-flap DNA substrate. To address this question, we created oligonucleotides containing the abasic (tetrahydrofuran, THF) site analog[29] along the 3′ (AP1–4) or 5′ (AP5–8) strand of a double-flap substrate (Fig. 4a). Although the Csm2-Psy3 heterodimer binds all of the DNA substrates as revealed by equilibrium-binding assays using a fluorescence polarization anisotropy technique, we observe an almost two-fold improved affinity (lower $K_d$) to a double-flap substrate containing an abasic site analog on the ssDNA at the ssDNA/dsDNA junction on the 5′ strand [AP6; $K_d = 71 \pm 7.3$ nM; Fig. 4b, c, Supplementary Table 2]. To determine if Csm2-Psy3 binding to AP6 is sequence dependent, we changed the nucleotide opposite of the abasic site analog and examined DNA binding by electrophoretic mobility shift assay (EMSA; Supplementary Fig. 5a). The preferential binding of Csm2-Psy3 to AP6, and the adjacent AP7, occurs independently of the nucleotide sequence (Supplementary Fig. 5a). It is interesting that we do not observe increased Csm2-Psy3 DNA binding with AP7 by anisotropy or EMSA. Furthermore, the different binding affinities between Csm2-Psy3 with AP6 and AP7 are not distinguishable by EMSA analysis, which is a qualitative and less sensitive assay. Note that the condition where we are able to visualize Csm2-Psy3 protein-DNA complexes in the acrylamide gel, and not in the wells, is in the presence of 5% glycerol. The addition of glycerol may stabilize and inhibit the dissociation of substrate from Csm2-Psy3-DNA complexes in the gel matrix in comparison to the quantitative equilibrium affinities obtained above.

Next, we determined whether Csm2-Psy3 would bind with improved affinity to a DNA structure that more closely resembles an in vivo replication fork with one or two dsDNA regions[30]. Using an EMSA, we find that Csm2-Psy3 binds to double-flap, 5′

generated by Ung1 glycosylase activity. Moreover, the *can1* mutations in the *csm2Δ* and *psy3Δ* strains maintained a G nucleotide strand-bias observed in wild type and exacerbated in *ung1Δ* cells (Fig. 3d), which is indicative of lagging strand-associated mutagenesis. Together these results indicate that the

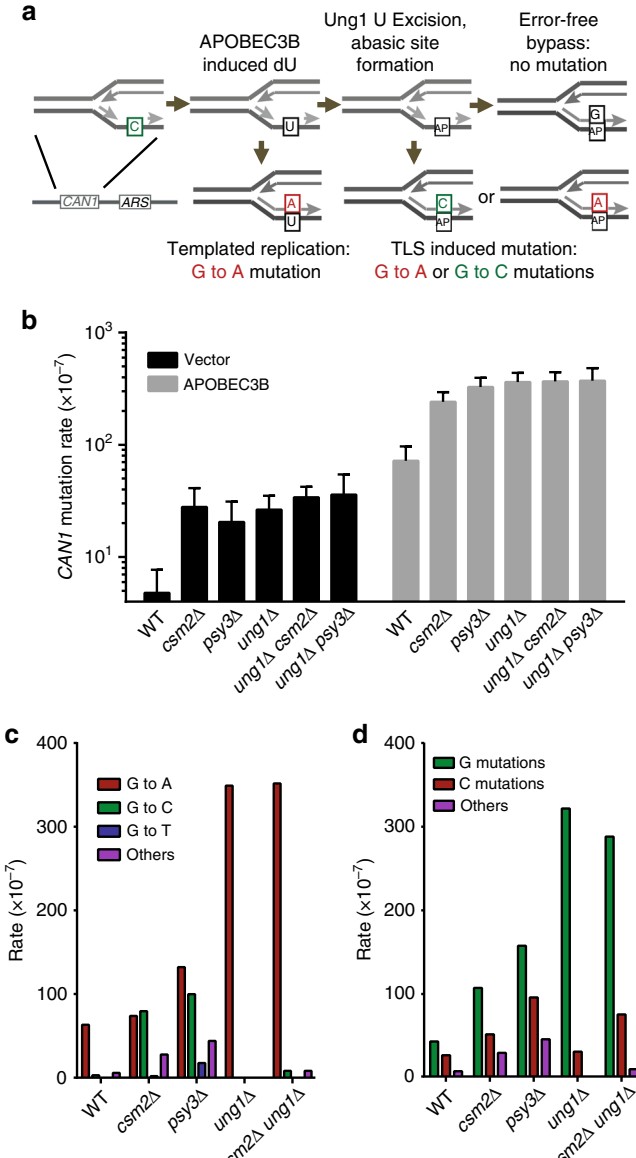

**Fig. 3** The Shu complex promotes bypass of APOBEC3B-induced lesions. **a** APOBEC3B-induced mutation rates on the lagging strand of a replication fork measured using a *CAN1* reporter integrated 16 kb from ARS216 on chromosome II. Expression of the cytidine deaminase APOBEC3B induces primarily lagging strand mutations caused by dU templated replication (G to A transition). The uracil glycosylase Ung1 removes U resulting in abasic site formation (AP) in the lagging strand, which can be bypassed by TLS (G to A transition, G to C transversion). **b** *CSM2* and *PSY3* are in the same pathway as *UNG1* and their deletion results in similar mutation rates individually or in combination with each other. Mutation rates of the indicated genotypes were measured in *CAN1* reporter strains transformed with either an empty or APOBEC3B-expressing vector. Error bars indicate 95% confidence intervals. **c** In the absence of *CSM2* or *PSY3*, abasic sites accumulate and TLS predominates resulting in primarily G to A transitions (red) or G to C transversions (green) within the *CAN1* locus. APOBEC3B expression in WT, *ung1Δ*, or *csm2Δ ung1Δ* cells primarily result in G to A transitions. The rate reported represents the proportion of the CanR mutants observed from sequencing multiplied by the mutation rate determined in **b**. G to T substitutions are indicated in blue. Other mutations consisting of rare substitutions at A:T base pairs, insertions, deletions, and complex events composed of multiple mutations are depicted in purple. **d** The strand bias of *CAN1* mutations from APOBEC3B expression was evaluated by Sanger sequencing. APOBEC3B expression results in a mutation bias in the lagging strand from templated replication of C deaminations. *CSM2*, *PSY3*, or *UNG1* deleted cells exhibit more G mutations (green) than C mutations (red). Other mutations as defined in **b** are indicated in purple. Individual mutation rates were calculated as in **c**. Statistical significance of strand bias in APOBEC3B-expressing strains was determined by a two-tailed G-test with $p < 0.05$ for all genotypes

flap, and 3′ flap substrates with similar affinities and to a static replication fork with decreased affinity (Supplementary Fig. 5b). These results are consistent with decreased Csm2-Psy3 binding affinity for dsDNA[19]. Furthermore, we find that the full Shu complex (Csm2-Psy3-Shu1-Shu2) recapitulates observed substrate-binding affinities for AP6 ($K_d = 72 \pm 17$ nM) and WT ($K_d = 174 \pm 22$ nM) double-flap DNA compared to Csm2-Psy3 alone (Fig. 4d). Consistent with a lack of Shu1-Shu2 DNA-binding activity[11], we find that Shu1-Shu2 do not bind to AP6 (Supplementary Fig. 6a). Similarly, we do not observe a qualitative change in DNA binding between Csm2-Psy3 and the full Shu complex by EMSA (Supplementary Fig. 6b). Together, these results demonstrate that Csm2-Psy3-Shu1-Shu2 bind double-flap DNA substrates containing an abasic site analog in vitro.

**Csm2-Psy3 protects abasic site analogs from APE1 cleavage.** AP endonucleases cleave DNA adjacent to an abasic site to generate a 5′ deoxyribophosphate (5′dRP). Human APE1 is capable of incising abasic sites in both double-stranded and to a lesser extent single-stranded DNA[31]. In the context of DNA

replication, abasic site cleavage would result in fork collapse and DSB formation. Since we find that Csm2-Psy3 binds to double-flap DNA structure containing an abasic site analog THF at the flap junction (AP6; Fig. 4a, c), we hypothesized that Csm2-Psy3 binding would inhibit AP endonuclease activity towards AP6. To test this hypothesis, we performed endonuclease assays using human APE1 with the AP6 substrate in the presence or absence of Csm2-Psy3. Indeed, we observe reduction of AP endonuclease cleavage of AP6 upon increasing concentration of Csm2-Psy3 protein (Fig. 5a). Using in vitro pull-down experiments, we find that Csm2-Psy3 inhibition of APE1 activity towards AP6 is unlikely due to a direct interaction between these proteins (Supplementary Fig. 7). The endonuclease assay was initially incubated for one minute to limit the number of APE1 catalytic cycles. However, we observe significantly reduced AP6 cleavage in the presence of Csm2-Psy3 even over an extended time course (Fig. 5b). APE1 cleavage is most effective on double-flap substrates where the abasic site analog is in the dsDNA region (AP8), consistent with previous findings ([32]; Supplementary Fig. 8a, b, compare AP6 to AP8). Therefore, Csm2-Psy3 attenuates AP endonuclease cleavage and DSB formation by binding to an abasic site at the double-flap junction.

We next determined whether the presence of Rad51 or Shu1-Shu2 would alter Csm2-Psy3 inhibition of AP6 cleavage by APE1. To do this, we determined that Rad51 binds to AP6, but with decreased affinity than Csm2-Psy3 and is saturated at 2 μM (Supplementary Fig. 9). We next tested whether Rad51 would block AP6 cleavage by APE1 and find that it does (Fig. 5c). We then examined whether Csm2-Psy3 together with Rad51 (2 μM) would further inhibit APE1 endonuclease activity. We find that the presence of both Rad51 and Csm2-Psy3 reduces APE1 endonuclease activity the most (Supplementary Fig. 8c). In contrast, Shu1-Shu2 does not significantly increase Csm2-Psy3 inhibition of APE1 cleavage in the presence of Rad51 (Fig. 5d).

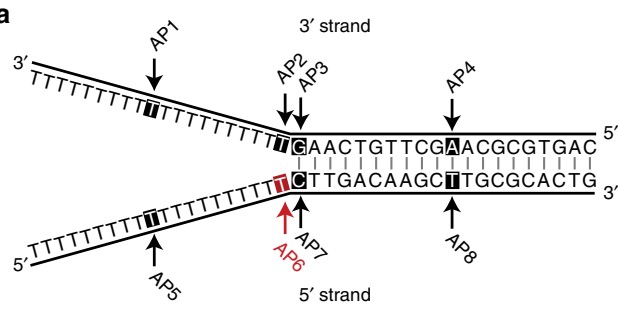

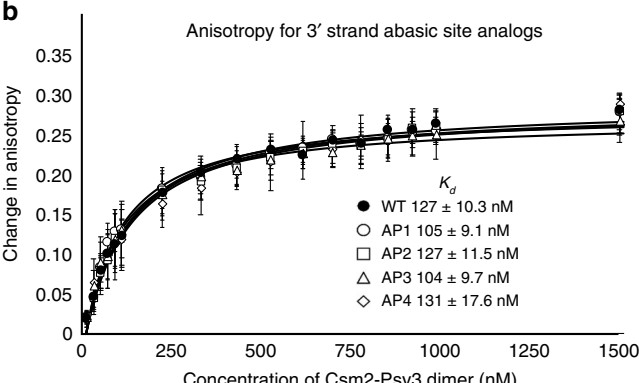

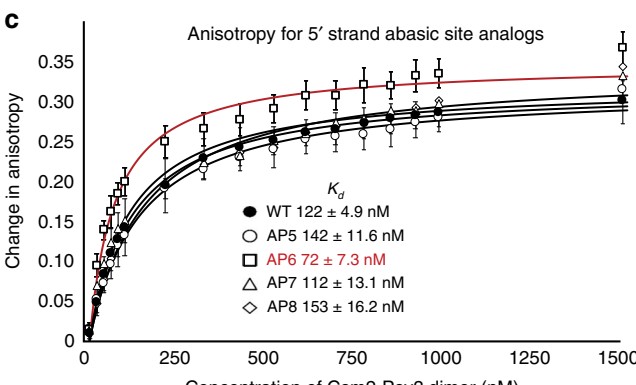

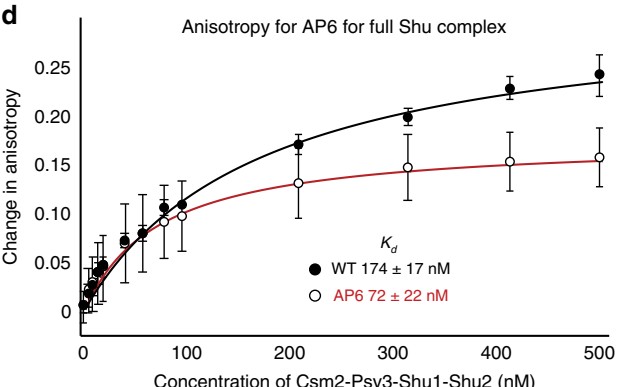

**Fig. 4** Csm2-Psy3 binds an abasic site analog in double-flap DNA. **a** Schematic of double-flap substrates containing abasic site analogs (tetrahydrofuran; indicated by an arrow and highlighted text) on the 3′ (AP1–4) or 5′ (AP5–8) DNA strand. Note that the complementing oligonucleotide, which is unmutated, is FITC labeled on the ssDNA end of the double-flap substrate. **b** Csm2-Psy3 binds to double-flap substrates with abasic site analogs along the 3′ DNA with equal affinity. DNA binding by Csm2-Psy3 was measured by fluorescence anisotropy, with increasing concentrations of the Csm2-Psy3 heterodimer added to 25 nM fluorescein-labeled double-flap substrate (AP1–4). Error bars indicate standard deviations measured in triplicate experiments. Data were fit to a one-site binding model and average dissociation constants ($K_d$) were calculated. Note the improved binding compared to Fig. 1 due to increased sample purity (described in detail in Supplementary Methods). **c** Csm2-Psy3 binds more tightly to a double-flap substrate containing an abasic site analog on the 5′ oligonucleotide at the junction (AP6, highlighted in red). In vitro-binding assays as described in **b** except with double-flap substrates containing 5′ abasic site analogs (AP5–8). **d** DNA binding by the Csm2-Psy3-Shu1-Shu2 was measured by fluorescence anisotropy, with increasing concentrations of the indicated proteins added to 5 nM AP6 (highlighted in red) or 2.5 nM double-flap DNA substrate. The data were fit to a quadratic equation

that the leading and lagging strands may be differentially recognized by specific DNA repair factors and targeted for repair through unique mechanisms. For example, the lagging strand contains more ssDNA regions, which inherently make it more prone to spontaneous damage as well as accessible to DNA-damaging agents. Here we propose that the Rad51 paralogs, Csm2-Psy3, directly recognize and tolerate abasic sites (Fig. 6). Rad51 paralog binding to abasic sites prevents AP endonuclease cleavage and potential formation of cytotoxic DSBs. This function is not unprecedented as RPA blocks APE1 cleavage of an abasic site analog on ssDNA and a double-flap substrate[33]. It is interesting to note that in mammalian cells, the HMCES protein forms protein–DNA crosslinks at abasic sites, shielding these sites from TLS or APE1-induced DSBs[34]. In contrast, Rad51 paralog binding to specific fork blocking lesions, such as an abasic site and perhaps other fork blocking lesions, would promote Rad51 filament formation enabling a template switch using the newly synthesized sister chromatid. By template switching, the lesion would be bypassed by the replication machinery in an error-free manner and could subsequently be repaired by BER after the fork progresses. At the same time, disruption of Shu complex ability to recognize and bind to abasic sites results in error-prone repair, such as TLS and single-strand annealing[14], to predominate.

Here we present extensive in vitro and in vivo evidence for a function of the Shu complex in tolerance of abasic sites. We show that (1) Shu complex member Csm2 chromatin association is enriched upon abasic site accumulation but not stalled forks (Fig. 2b, c, Supplementary Fig. 4b); (2) Csm2 DNA binding is required for its chromatin association when abasic sites accumulate and these mutants exhibit extreme DNA damage sensitivity and are mutagenic (Fig. 2a, b); (3) csm2Δ and psy3Δ mutants exhibit mutation signatures consistent with abasic site repair on the lagging strand (Fig. 3); (4) Csm2-Psy3 and Csm2-Psy3-Shu1-Shu2 bind with improved affinity to a double-flap substrate containing an abasic site analog (THF) at the junction (AP6; Fig. 4c, d); and lastly (5) Csm2-Psy3 protects AP6 double-flap substrates from in vitro endonuclease cleavage (Fig. 5). One interesting aspect of this study is the twofold improved affinity observed of Csm2-Psy3 for AP6 but not AP7 (Fig. 4b, c). It is possible that there is a binding pocket in the Csm2-Psy3 complex that can accommodate an abasic site analog when it is only in the

Together these results suggest that both Rad51 and Csm2-Psy3-Shu1-Shu2 are capable of blocking APE1 endonuclease activity.

## Discussion

DNA damage can arise from many different sources and damage that is encountered by the replication fork can result in fork stalling, collapse, and DSB formation (Fig. 6). Our results suggest

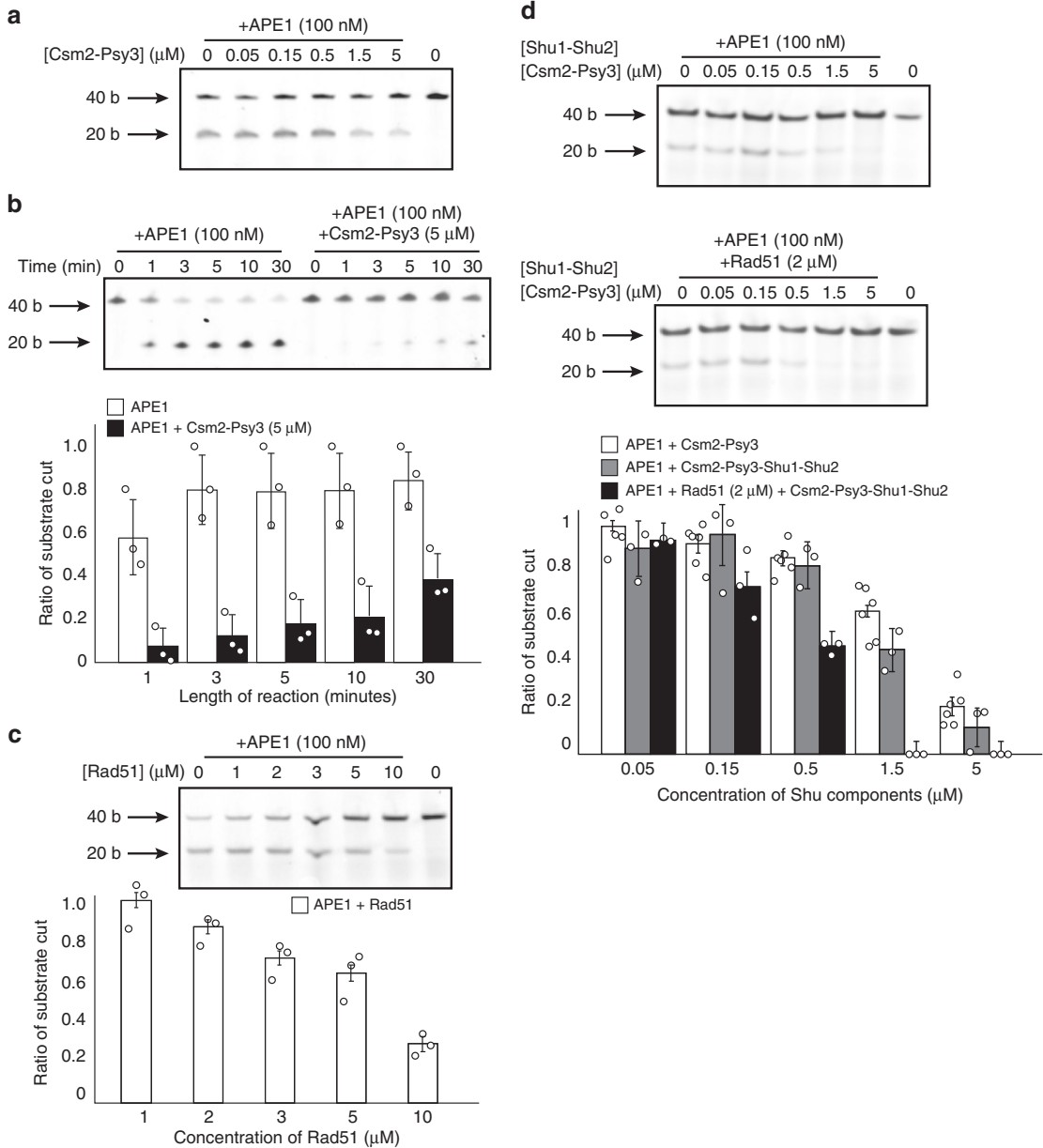

**Fig. 5** APE1 activity is inhibited by Csm2-Psy3 DNA binding. **a** Increasing Csm2-Psy3 protein concentrations decrease APE1 activity on a double-flap substrate containing a lagging strand abasic site analog (AP6). Representative gel and bar graph showing dose-dependent protection of the double-flap abasic site analog at the junction (AP6, 100 nM) by Csm2-Psy3 (0–5 μM) from APE1 AP endonuclease activity (100 nM). AP6 was incubated with Csm2-Psy3 or buffer for 5 min before APE1 addition. Reactions were stopped after 1min to limit APE1 catalytic cycles. Error bars indicate standard deviations from three experiments. **b** Csm2-Psy3 protects a double-flap substrate containing a 3′ abasic site analog (AP6) over time from AP endonuclease cleavage. Representative gel and bar graph showing Csm2-Psy3 (5 μM) protection of AP6 (100 nM) from APE1 endonuclease activity (100 nM) over an extended time course (0–30 min). More than 60% of AP6 bound by Csm2-Psy3 persists even after 30min. Error bars represent standard deviations from three experiments. **c** Rad51 inhibits APE1 cleavage of AP6 in a concentration-dependent manner. Shown is a representative gel from three experiments. Results from three experiments are quantified and error bars represent standard deviations. **d** Shu1-Shu2 does not significantly enhance Csm2-Psy3 protection of AP6 against APE1 endonuclease. Representative gels and bar graph showing protection of AP6 (100 nM) against APE1 (100 nM) by Shu1-Shu2-Csm2-Psy3 (0–5 μM) with (bottom gel) or without (top gel) added Rad51 (2 μM). Error bars represent standard deviations from three experiments

AP6 position compared to the AP7 position. The nucleotide adjacent to the abasic site analog may also alter the DNA structure and therefore influence DNA binding activity of Csm2-Psy3[29,35]. In addition, it remains unknown how Rad51 and Rad55-Rad57, which directly interact with Csm2-Psy3, may contribute to this substrate specificity. Future atomic resolution studies will be necessary to understand the specificity differences between these two substrates. Together, the combined in vitro and in vivo complementary data described above provide the

strongest evidence that the Shu complex has an important role in tolerance of abasic sites.

Our in vitro findings suggest a role for the Shu complex in preventing DSB formation during replication and further studies are needed to demonstrate that DSBs are indeed increased in vivo upon Shu complex disruption. However, consistent with increased DSB formation, Shu complex mutant cells exhibit more Rad52 foci upon MMS exposure in S/G2/M cells compared to wild type[10] and a delay in chromosome reconstitution upon

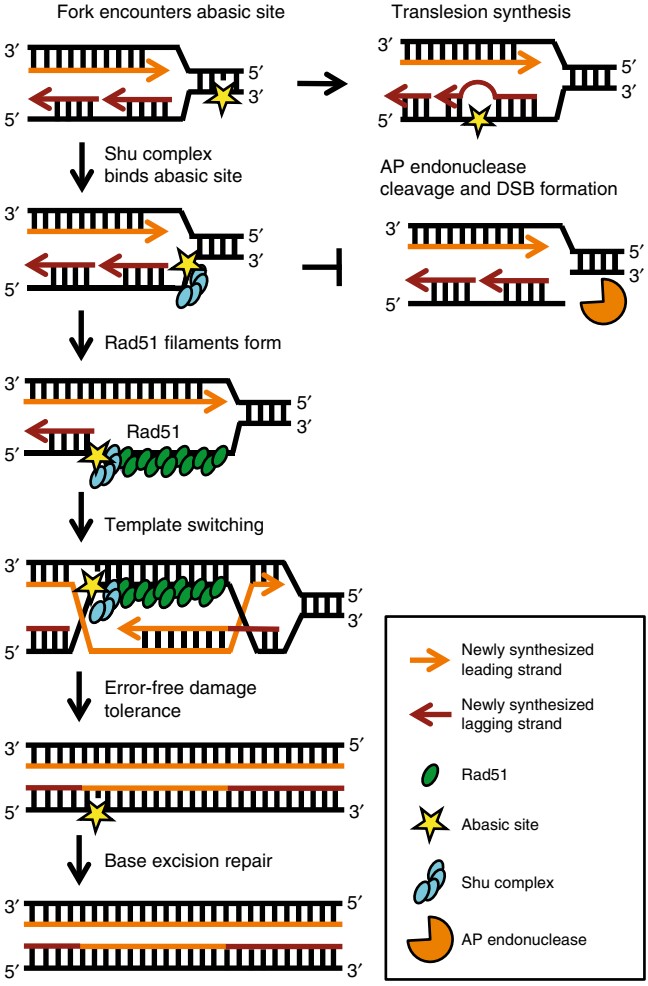

**Fig. 6** Model of Shu-mediated DNA strand-specific damage tolerance. The Shu complex DNA-binding components, the Rad51 paralogs Csm2-Psy3, bind to abasic sites at a double-flap junction to promote Rad51-mediated template switching while preventing AP endonuclease cleavage. MMS-induced DNA damage is primarily repaired by the BER pathway. However, if a replication fork encounters DNA damage such as an abasic site (yellow star), then the fork can stall or collapse. The Shu complex (blue ovals) binds abasic sites on the lagging strand template proximal to the dsDNA fork stem. Shu complex DNA binding (1) promotes Rad51 filament formation (green ovals) and (2) likely prevents AP endonuclease cleavage (orange Pac-man) and DSB formation. Thus, the Shu complex mediates a DNA strand-specific damage tolerance pathway enabling error-free lesion bypass through a template switch using the newly synthesized sister chromatid. This strand-specific lesion bypass pathway allows replication to continue efficiently in an error-free manner and the abasic site to be repaired by BER after the fork progresses

MMS exposure in S phase synchronized culture[14]. It is interesting to note that *csm2Δ* in combination with accumulation of abasic sites (from an *apn1Δ apn2Δ ntg1Δ ntg2Δ* mutant), results in 1075× increase in spontaneous mutation rates[14], which likely accounts for the extreme MMS sensitivity observed in this mutant background.

Although we find that the Shu complex exhibits improved binding affinity for double-flap structures, we previously showed that the Csm2-Psy3 heterodimer also binds to 5′ and 3′ DNA overhangs[19]. In this context, the Shu complex could bind to a 5′ overhang that forms when a replicative polymerase stalls at a DNA lesion and then dissociates from it. It is also possible that other DNA repair factors or the replication machinery itself may

also contribute to Shu complex recruitment to DNA damage at a replication fork. In this scenario, the Shu complex could recruit Rad51 to stalled replication forks to facilitate a template switch.

Our work has important clinical implications as the interdependency between DNA repair pathways needed during replication is being exploited for cancer treatment. For example, human BRCA1 and BRCA2 function during HR by promoting resection and RAD51 activity, respectively[36]. BRCA1 and BRCA2 disruption is associated with hereditary breast and ovarian cancers. PARP inhibitors are effective in the treatment of patients with BRCA1- and BRCA2-deficient tumors[37–39]. Recent studies have extended these observations and PARP inhibitors are now being used to treat patients with RAD51 paralog-deficient tumors in clinical trials[40]. Therefore, upon replication stress when early DNA repair steps are blocked by PARP inhibition, HR is required to bypass the lesion. A recent study has implicated PARP1 in ligation of Okazaki fragments where PARP inhibition prevents Okazaki fragment ligation, which would then require HR for removal[41]. In this replicative context, combined PARP inhibition with HR deficiency (due to BRCA or RAD51 paralog mutation) results in tumor cell lethality. Understanding the underlining mechanisms of how BER intermediates, such as ssDNA breaks and abasic sites, are recognized and channeled for repair through HR is critical for exploiting DNA repair interdependency in cancer therapy to ensure the most durable clinical response.

## Methods

**Strains and plasmids.** The strains and plasmids used are listed in Supplementary Table 3. All strains are isogenic with W303 RAD5+W1588-4C[42] and W5059-1B[43] with the exception of the APOBEC3B-mediated mutagenesis assay. In the APOBEC3B-mediated mutagenesis, the yeast strains used for determining *CAN1* mutation rates and spectra are derived from wild-type ySR128[26], in which the *CAN1* gene is integrated approximately 16 kb to the left of ARS216. Construction of ySR128-derived *ung1Δ*, *mph1Δ*, and *ubc13Δ* strains was described[26,28]. HygMX cassettes for creating deletions of *CSM2* and *PSY3* were generated by PCR using primers described in Supplementary Table 4 and plasmid template pAG32 (ref. [44]). After transformation and selection, gene deletions were confirmed by PCR using primers that flank each gene (Supplementary Table 4). APOBEC3B expressing and empty plasmids were created in the vectors pySR-419 and pSR-440 (ref. [26]). A fragment containing the *LEU2* gene from pUG73 (ref. [45]) was PCR amplified using oligos oTM-74 and oTM-75. This fragment was then ligated into backbones created from PCR amplification of pySR-419 and pSR-440 with primers oTM-80 and oTM-81. The resulting plasmids, pTM-19 (empty vector) and pTM-21 (APOBEC3B expression vector), were validated by Sanger sequencing. The primers used for cloning and sequencing are found in Supplementary Table 4.

**Cananvanine mutagenesis assay.** Five individual *CAN1* colonies of WT, *csm2Δ*, *psy3Δ*, *csm2-KRRR*, *psy3-KRK*, and *csm2-KRRR psy3-KRK* were grown 2 mL YPD or YPD medium containing 0.00033% MMS (18 h) overnight at 30 °C. The cultures were diluted 1:10, with 250 μL plated on SC-ARG + CAN or diluted 1:60,000, with 120 μL plated on SC. The plates were then incubated for 48 h at 30 °C. Colonies were used to measure total cell number (SC) or forward mutation rates (SC-ARG + CAN). For each condition, colony count from at least four independent trails was used to calculate a mutation rate using a web-based calculator (http://www.mitochondria.org/protocols/FALCOR.htmL)[46] and the Lea-Coulson method of the median.

**APOBEC3B-mediated mutation rate and mutation spectra.** Yeast strains were transformed with either pTM-19 (empty vector) or pTM-21 (APOBEC3B expression plasmid) and selected on synthetic complete medium lacking leucine (SC-leu). Individual isolates were plated on SC-leu at a density of approximately 50 cells per plate and grown to colony sizes of $7 \times 10^7$ cells (for empty vector) or $7 \times 10^6$ cells (for APOBEC3B expression plasmid). Eight independent colonies were then resuspended in water and plated on SC or SC-arginine medium supplemented with 0.006% canavanine (SC + can) and incubated for 3 days at 30 °C. Colonies were used to measure total cell number (SC) or forward mutation rate (SC-ARG + CAN). For each condition, data from at least three independent transformants were used to calculate a mutation rate using a web-based calculator (http://www.mitochondria.org/protocols/FALCOR.htmL)[46] and the Lea-Coulson method of the median. Mutation spectra were determined for APOBEC3B-expressing strains plated on SC-leu medium at a density of about 50 cells per plate and grown until colonies reached approximately $7 \times 10^6$ cells. The resulting colonies were replica-plated to SC + can medium and grown for 4 days. To isolate independent clonal CanR mutants, papillae derived from discrete colonies were struck to single

colonies on SC + can medium and after 3 days one colony from each was patched onto YPDA medium. Genomic DNA was isolated from each patch and used as a template for amplification of the *CAN1* gene by PCR using primers oTM-92 and oTM-93 (Supplementary Table 4). The resulting PCR products were Sanger sequenced (GenScript, Piscataway, NJ), using primers oTM-94, oTM-95, and seqDG-91 (Supplementary Table 4) and the mutations inactivating *CAN1* were identified using the Geneious software package (Biomatters).

**Growth assays**. Yeast strains were incubated in 5 mL YPD medium overnight at 30 °C and then diluted to 5 mL OD$_{600}$ 0.2. The cultures were incubated for another 3 h at 30 °C to reach log phase, and 5 µL of culture at OD$_{600}$ 0.2 were fivefold serially diluted onto YPD medium or YPD medium containing 0.02% MMS. The plates were incubated at 30 °C for 2 days and photographed. Cell viability assays were performed by growing the indicated strains in 3 mL of YPD at 30 °C overnight, and then diluting the culture to 0.2 OD$_{600}$ for 3–4 h. The cultures were all diluted to 0.5 OD$_{600}$ in 1 mL YPD and diluted either 1:10,000 or 1:20,000 and 250 µL was plated onto YPD medium or YPD medium containing 0.012%, 0.02%, 0.03% MMS. The plates were incubated at 30 °C for 2 days before being counted. Representative images were taken after 2 days of growth at 30 ˚C°C for one of the experiments and the brightness and contrast was adjusted using Photoshop (Adobe Systems Incorporated). The experiment was performed five times with standard deviations calculated.

**Western blot analysis**. Five milliliters YPD was inoculated with the indicated cells and grown overnight at 30 °C. The cells were diluted to OD$_{600}$ 0.2 in 5 mL YPD and grown for 3 h at 30 °C. Whole-cell lysates of equal cell numbers (0.5 OD$_{600}$) were prepared by TCA precipitation[47] and 10 of 50 µL lysate preparation was run on a 10% SDS-PAGE gel where HA antibodies (sc-805; 1:500) were used to detect the 6HA tagged Csm2 and Psy3 proteins and Kar2 antibodies (Santa Cruz sc-33630; 1:200) were used as a loading control. The films were scanned and adjusted for contrast and brightness using Photoshop (Adobe Systems Incorporated). Unprocessed and uncropped film images are in source data file.

**Yeast-two-hybrids**. The yeast two-hybrid plasmid pGAD was used to express a fusion of GAL4-activation domain and pGBD was used to express a fusion of the GAL4 DNA-binding domain. The pGAD and pGBD indicated plasmids were transformed into PJ69-4A[2], and positive colonies were selected on synthetic complete without tryptophane and leucine (SC-TRP-LEU) medium. Individual colonies were grown to early log phase (OD$_{600}$ 0.2), and then 5 µL was spotted onto medium to select for the plasmids (SC-LEU-TRP) or onto medium to select for expression of the reporter *HIS3* gene (SC-LEU-TRP-HIS), indicating a yeast two-hybrid interaction. Plates were incubated for 2 days at 30 °C and subsequently photographed. Each experiment was done in triplicate.

**Protein expression and purification**. All Csm2-Psy3 heterodimers were cloned into the dual expression plasmid pRSFDuet (EMD Millipore) which encode a 6XHIS-TEV tag on Csm2. Transformed *E. coli* [BL21-Codon+ (DE3, RIL) Agilent] was grown at 37 °C to 0.6 OD$_{600}$ and recombinant protein expression was induced by addition of 0.2 mM isopropyl beta thiogalactoside (IPTG) at 18 °C overnight for 16–18 h. Cells were harvested by centrifugation. Approximately 10 g of cell pellet was lysed in 60 mL of lysis buffer containing 20 mM Tris (pH 8.0), 500 mM NaCl, 10% glycerol, 5 mM imidazole, and 1 mM β-mercaptoethanol supplemented with protease inhibitors (Roche) and DNAse (1 µg mL$^{-1}$). Cells were lysed using an emulsiflex and centrifuged at 30,000 × *g* for 1 h at 4 °C. Csm2 and Psy3 were co-purified through nickel affinity chromatography (Qiagen) using the N-terminal His$_6$-tag on Csm2 in Nickel binding buffer (20 mM Tris pH 8.0, 500 mM NaCl, 10 mM Imidazole, and 1 mM beta-mercaptoethanol). Csm2-Psy3 was washed on the column with 50 mL of binding buffer containing 10, 15, and 20 mM imidazole to remove contaminating proteins. The Csm2-Psy3 was eluted from the column with elution buffer containing 20 mM Tris pH = 8.0, 500 mM NaCl, and 250 mM Imidazole. Wild-type Csm2-Psy3 dimers used in the abasic binding experiments were further purified using HiTrap Heparin HP (GE Healthcare) affinity chromatography. The Csm2-Psy3 protein was loaded onto the heparin column equilibrated in buffer containing Tris pH 8.0, 1 mM beta-mercaptoethanol, and 8% glycerol. The complex was eluted with a gradient elution from 25% to 100% (Tris pH 8.0, 1 M NaCl, 1 mM beta-mercaptoethanol, and 8% glycerol) over 75 mL. The Csm2-Psy3 protein typically eluted around 400–600 mM NaCl. Since mutant Csm2-Psy3 dimers fail to bind the heparin column, wild-type Csm2-Psy3 and mutant Csm2-Psy3 constructs were purified using a HiTrap Q (GE Healthcare) anion exchange column for a direct comparison of DNA-binding affinities. Note that this change in purification scheme results in different binding affinities compared to the protein preparations that used the heparin column. All Csm2-Psy3 constructs were subsequently purified by size exclusion chromatography using a Sephacryl S200 column (GE Healthcare) in buffer (Tris pH 8.0, 1 M NaCl, 1 mM beta-mercaptoethanol, and 8% glycerol), eluting as a single peak (Supplementary Fig. 3d) and visualized as heterodimers by SDS-PAGE electrophoresis (Supplementary Fig. 1). Csm2-Psy3 protein concentration was determined by absorbance at $A_{280}$ with an extinction coefficient of 54,320 M$^{-1}$ cm$^{-1}$. For full description of Shu1 and MBP-Shu2 purification see ref. [48]. Briefly, Shu1 and MBP-

Shu2 were expressed in *E. coli* (Rosetta [DE3]) by transforming cells with the pET-DUET vector encoding 6XHIS-Shu1 and MBP-tagged Shu2 (tags located on the N-terminus). Cells were grown in 2× LB broth containing 0.1 mM ZnCl$_2$ at 37 °C until OD$_{600}$ 0.8 was reached, recombinant protein expression was induced with a final concentration of 0.2 mM IPTG, and shifted to grow at 16 °C for 16 h. Cells were harvested by centrifugation and 40 g of pellet was resuspended in 200 mL of buffer with 300 mM KCL containing protease inhibitors. Cells were lysed by sonication and centrifuged at 100,000*g* for 1 h at 4 °C. Then supernatant was incubated with Ni-NTA resin, washed with buffer containing 150 mM KCl and 10 mM imidazole, and eluted by 200 mM Imidazole. Eluate was incubated for 2 h with amylose resin with gentle mixing and eluted. Elution was run on a SuperDex 200 column in buffer with 150 mM KCl buffer. Protein eluted as a monodispersed dimeric protein complex. Peak fractions were pooled, concentrated in an Amicon Ultra micro-concentrator, snap frozen in liquid nitrogen, and stored at −80 °C. For description of full purification method of Rad51 see ref. [49]. Unprocessed gel images are in source data file.

**Equilibrium-binding assays using FPA**. Anisotropy experiments were performed using a FluoroMax-3 spectrofluorometer (HORIBA Scientific) and a Cary Eclipse Spectrophotometer. For unmodified forks, fluorescein dT was incorporated at the 5′ single-stranded end of the fork (Fig. 1b). For double-flap substrates containing abasic site analogs, the label was placed on the single-stranded end of the oligo-nucleotide that did not contain the abasic site analog. Anisotropy measurements were recorded in a 500 µL cuvette containing 20 mM Tris pH 8.0 and 20 nM of fluorescein-labeled double-flap substrate as a premixed sample of purified Csm2-Psy3 protein and substrate (20 nM AP6 + 1.6 µM Csm2-Psy3 dimer in 1 M NaCl) was titrated into the cuvette. Fluorescence anisotropy measurements were recorded using the integrated polarizer and excitation and emission wavelengths of 466 and 512 nm, respectively, with path lengths of 10 nm. Titrations were carried out until anisotropy became unchanged. At the end of each titration, the DNA substrate was competed off with 1 M NaCl to confirm that the increase in anisotropy was explained by bona fide electrostatic interactions with Csm2-Psy3. All experiments were performed in triplicate with multiple preparations of the recombinant proteins. Dissociation constants ($K_d$) were calculated by fitting our data to a one-site binding model using the equation for a rectangular hyperbola [$Y = B_{max}*X/(K_d + X)$], with PRISM7 software (Supplementary Tables 1 and 2). Anisotropy experiments with 2.5 and 5 nM substrate concentrations were fit to a quadratic equation [$Y = M* ((x + D + K_d) - sqrt(((x + D + K_d)^2) - (4*D*x))/(2*D)$], with PRISM7 software. Unprocessed raw anisotropy values are in the source data file.

**Electrophoretic mobility shift assay**. EMSA reactions were performed in buffer containing 25 mM Tris-HCl (pH 8.0), 50 mM NaCl, 1 mM DTT, 100 µg mL$^{-1}$ bovine serum albumin (BSA) (Sigma), and 5% glycerol. Briefly, FITC-annealed substrates (25 nM) or Cy3-annealed substrates (5 nM) were incubated with increasing concentrations of Csm2-Psy3, Shu1-Shu2, or Csm2-Psy3-Shu1-Shu2 (0, 50, 100, 200, 400, 600, and 800 nM) at 25 °C for 5 min. At the end of the reaction, 0.4 µL of an NP40 and d*I*/d*C* were added to a final concentration of 0.23% and 7.7 ng mL$^{-1}$, respectively. Samples were run on a 5% TBE (pH 8.5) acrylamide gel containing 5% glycerol at 4 °C for 1 h at 70 V. Gels were imaged on a Typhoon 9400 Variable Mode Imager (GE Healthcare) with an excitation wavelength of 488 nm and emission wavelength of 526 nm with a PMTV at 650. Free or bound substrate intensities were quantified using ImageQuant TL 1D v8.1 software. Quantification of Csm2-Psy3 bound to the different flap substrates was performed by dividing the % bound complex by the total substrate intensity in each lane. Error bars represent standard deviations. Unprocessed and uncropped typhoon images are in source data file.

**Chromatin fractionation**. Chromatin fractionation was based on refs. [50,51] with modification. Five milliliters YPD was inoculated with the indicated cells and grown overnight at 30°C. The cells were diluted to 0.2 OD$_{600}$ in 50 mL YPD and grown for 3 h at 30C. The cells were then diluted to 0.3 OD$_{600}$ in 50 mL fresh YDP with 20 µM α-factor (GeneScript). After 2 h incubation at 30 °C, the cells were pelleted and washed with 50 mL YPD. The culture was diluted to 0.5 OD$_{600}$ in 50 mL fresh YPD or YPD containing the indicated MMS concentration (0.01%, 0.02%, or 0.03%) or HU concentration (50 or 200 mM). After 1 h incubation at 30°C, 30 OD$_{600}$ cells were washed with 50 mL ice-cold water and resuspended in 2 mL pre-spheroplast buffer (100 mM PIPES/KOH pH 9.4, 10 mM DDT, 0.1% NaN$_3$)[50] for 10 min at room temperature. The cells were pelleted and then resuspended in 3 mL spheroplast buffer (50 mM K$_2$HPO$_4$/KH$_2$PO$_4$ pH 7.5, 0.6 M Sorbitol, 10 mM DTT, 0.1 µg mL$^{-1}$ Zymolyase 100T [amsbio])[50] and incubated for 40 min at 30°C (120 rpm). The spheroplasts were pelleted and washed with ice-cold wash buffer (50 mM HEPES/KOH pH 7.5, 100 mM KCl, 2.5 mM MgCl$_2$, 0.4 M sorbitol)[50]. The spheroplasts were pelleted and resuspended in 80 µL of extraction buffer (wash buffer with 1 % Triton X-100, 1 mM DTT, protease inhibitors, and 2 mM PMSF)[51]. The spheroplasts were lysed by vortexing for 5 min with intermittent incubation on ice. Eighty microliters of HU loading buffer (8 M urea, 5% SDS, 200 mM Tris pH 6.8, 1 mM EDTA, 0.02% w/v bromophenol blue, 0.2 M DTT)[47] was added to 20 µL of each lysate and set aside for analysis [whole-cell lysate (W)]. The remaining lysate was loaded on top of a 50 µL sucrose cushion

(wash buffer with 30% sucrose, 0.25% triton X-100, 1 mM DTT, protease inhibitors, and 2 mM PMSF)[51] and centrifuged 10 min at 4 °C at 20,000 × g. Eighty microliters of HU loading buffer was added to 20 µL from the top layer and set aside for analysis [non-chromatin fraction (S)]. The chromatin fraction (C) pellets were resuspended in 100 µL of HU loading buffer. Five microliters of the W, S, and C samples were run on a 12% SDS-PAGE gel and western blot analysis was performed. HA antibodies (sc-805; 1:500) were used to detect the 6HA-tagged Csm2 protein, Kar2 antibodies (Santa Cruz sc-33630; 1:200), GAPDH (UBPbio Y1040; 1:10000), Rfa1 (Abcam ab221198; 1:6000), and H2B antibodies (Active Motif #39237, 1:1000) were used for controls. The films were scanned and adjusted for contrast and brightness using Photoshop (Adobe Systems Incorporated). Unprocessed and uncropped film images are in source data file.

**Chromatin fractionation data analysis.** Each experimental condition was repeated 3–5 times. The densitometry analysis was performed using ImageJ software[52] to quantify the whole-cell lysate (W), non-chromatin fraction (S), and chromatin fraction (C). To analyze the amount of Csm2 that is chromatin associated, the signal of Csm2 in the C fraction was divided by the W fraction. Kar2 chromatin association was also calculated by dividing Kar2 C fraction by the W fraction. To account for chromatin extraction efficiency, the calculated Csm2 chromatin association value was then divided by the corresponding Kar2 chromatin association value. Finally, to account for loading differences, we compared the Csm2 and Kar2 W fractions. To compare Csm2 chromatin enrichment between experiments and obtain fold changes, we set the untreated Csm2-6xHA strain chromatin signal to 1 and the averages of each trial were plotted with standard deviations and significance determined by unpaired two-tailed Student's $t$-test. Unprocessed and uncropped film images are in source data file.

**Synthesis of oligonucleotides containing an abasic residue.** Standard DNA phosphoramidites with labile phenoxyacetyl (PAC), acetyl (Ac), or isopropyl-phenoxyacetyl (iPrPAC) protecting groups, dSpacer phosphoramidite, deoxythymidine CPG, deoxycytidine CPG, and standard solid phase synthesis reagents were purchased from Glen Research (Sterling, VA). Solid phase synthesis was performed on a Mermade-4 (Bioautomation, Plano, TX, USA) automated synthesizer. DNA synthesis and deprotection were performed using standard protocols following the manufacturer's recommendations. After deprotection, DNA was analyzed for purity using reverse phase HPLC. The HPLC system consisted of a Waters 1525 pump system and a Waters 2998 photodiode array detector using a Waters XBridge column (C18 130 Å) in 0.1 M triethylamine acetate and 80:20 acetonitrile:water in 0.1 M triethylamine acetate at 25 °C. Strands showing significant impurity profiles were purified by either HPLC or gel if necessary. The DNA band was excised and eluted overnight in $TE_{0.1}$ buffer (10 mM Tris, 0.1 mM EDTA, pH 7.5). Eluted DNA was desalted using a C18 Sep-Pack cartridge (Waters, Milford, MA, USA). All DNA was characterized by MALDI-TOF mass spectrometry using a 3-hydroxypicolinic acid matrix. The sequence of the abasic site analog containing DNA is shown in Supplementary Table 5. The sequences of the Cy3 labeled double-flap, 5′ flap, 3′ flap, and static replication fork are shown in Supplementary Table 6 and were adapted from ref. [30].

**Pull-down assays.** Pull-down assays were performed by incubating 25 µL Ni beads (Qiagen) with 1 µM of APE1 and 1 µM Csm2-Psy3 in 20 mM Tris (pH 8.0), 50 mM NaCl, 5 mM MgOAc₂, 5% glycerol, 1 mM DTT, 2 mM ATP, and 0.5 mg mL⁻¹ BSA to a final volume of 30 µL. The reaction was incubated over night at 4°C with rotation and then subsequently pelleted at 3000 × g for 5 min. Unbound sample was saved for gel analysis and beads were washed twice with 500 µL of buffer and bound proteins were eluted with 5 µL of binding buffer and 5 µL of SDS-PAGE loading buffer (250 mM Tris-Cl pH 6.8, 8% SDS, 0.2% bromophenol blue, 40% glycerol, and 20% beta-mercaptoethanol) followed by boiling for 4 min. Proteins were separated on a 15% SDS-PAGE gel for analysis. Gel image was adjusted for contrast. Unprocessed gel images are in source data file.

**APE1 endonuclease activity assay.** All endonuclease experiments were performed in vitro using recombinant components. One hundred nanomolar of 3′ fluorescein-labeled double-flap substrate was dissolved in reaction buffer containing 20 mM HEPES (pH 7.5), 50 mM NaCl, 1 mM MgCl₂, 5% glycerol, 1 mM DTT, and 0.5 mg mL⁻¹ BSA. DNA substrate and Csm2-Psy3 were incubated for 5 min before adding human APE1 (ref. [53]) (generously provided as a gift from Sam Wilson, NIEHS and New England BioLabs) or buffer. For the titration experiments, the reaction mix contained AP6 and either buffer alone or dilutions of Csm2-Psy3 ranging from 50 nM to 5 µM. One hundred nanomolar APE1 was then added and the reaction mixes were incubated at room temperature for 1 min and stopped by adding an equal volume of formamide loading buffer (95% formamide, 5 mM EDTA) and boiling for 5 min. Samples were then resolved by running on 8% TBE-urea gels at 35 mA for 8 min. Gels were analyzed using the image processing software ImageJ[52]. Relative enzyme activity was determined by measuring the ratio of uncut to cut APE1 products in each lane. Results for each condition were averaged across experimental triplicates and then normalized against a control APE1 condition without added Csm2-Psy3. The time course assay was performed similarly with the following modifications: samples contained fixed amounts of APE1 (100 nM) and Csm2-Psy3 (5 µM). Reactions were then incubated at room temperature for 30 min, with samples collected at intervening time points. Reactions containing Rad51 were performed as described for Csm2-Psy3 with the following modifications: Reaction buffer for all Rad51-involved assays contained 20 mM Tris (pH 8.0), 50 mM NaCl, 5 mM MgOAc₂, 5% glycerol, 1 mM DTT, 2 mM ATP, and 0.5 mg mL⁻¹ BSA. Samples were incubated as described for Csm2-Psy3 before adding human APE1 or buffer. Titrations contained dilutions of Rad51 ranging from 1 to 10 µM for the Rad51 inhibition assays. For assays measuring the combined effect of Rad51 and Csm2-Psy3, titrations of Csm2-Psy3 ranging from 50 nM to 5 µM were supplemented by the addition of 2 µM Rad51. Titrations involving the entire Shu complex (Csm2-Psy3-Shu1-Shu2) were performed exactly as described for Csm2-Psy3, with the concentrations of Shu1-Shu2 equimolar to Csm2-Psy3. Quantification, averaging, and normalization for all experiments were performed as described for Csm2-Psy3. Gels were contrast adjusted and unprocessed and uncropped film images are available in the source data file.

## Data availability

The authors declare that the source data (Fig. 1b, d, Fig. 2c, d, Fig. 3b–d, Fig. 4b–d, Fig. 5a–d, Supplemental Fig. 1, Supplemental Fig. 2a, b, Supplemental Fig. 3a–c Fig. 4a, b, Fig. 5a, b, Fig. 6a, b, Fig. 7a, Fig. 8a–c, and Fig. 9) support the findings in this study and are available within the paper, within the supplementary files, and all data are provided as a Source Data file. All data are also available from the authors upon reasonable request.

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

## Acknowledgements

We thank Samuel Wilson for the APE1 enzyme. We thank William Gaines for assisting with the protein purification. This study was supported by the National Institutes of Health grant (ES024872 to K.A.B. and A.P.V.; ES019566 to B.V.H.; ES007061 to P.S.; CA218112 to S.A.R.) and the American Cancer Society (129182-RSG-16-043-01-DMC to K.A.B.). K.P. is supported by a Starting grant from the European Research Council (ERC Stg Grant: 638988-G4DSB). This project used the UPMC Hillman Cancer Center Biostatistics Facility that is supported in part by award P30CA047904.

## Author contributions

J.C.R., B.B., S.R.H., B.W.H., A.D.M., T.M.M., H.G.K., C.A.P., E.M. performed the experiments. J.C.R. and S.R.H purified recombinant proteins and performed biochemical analysis. T.C.R. created the abasic site analog substrates. S.K.G. created the DNA binding mutant yeast strains and plasmids. P.S. and Y.K. provided Shu1-Shu2 and Rad51 constructs and protein as well as technical assistance for experiments and purification strategies. D.P.N. did the statistical analysis for Fig. 1d. B.V.H., S.R.D., K.P., S.A.R., A.P.V., and K.A.B. designed the experiments. B.V.H, T.M.M., J.C.R., B.B., S.R.H., S.A.R., A.P.V., and K.A.B. wrote the manuscript.

## Additional information

**Competing interests:** The authors declare no competing interests.

