## [Peer Review File · Nature Communications]

Reviewers' comments:

Reviewer #1 (Remarks to the Author):

NCOMMS-18-14750-T

Rosenbaum et al.

The paper by Rosenbaum et al. describe the role of the yeast Shu complex, whose major subunits contains Psy3 and Csm2, Rad51 paralogs, in the repair of stalled replication forks at abasic (apurinic and apyrimidic; AP) sites. The authors identified critical residues for DNA binding activity of Csm2-Psy3 dimers. Importantly, using Chromatin immunoprecipitation (ChIP), Csm2 is shown to bind to replication origins, ARS. Purified Csm2-Pys3 inhibits an activity of human AP endonuclease on Y-fork DNA substrates containing an abasic site at the junction. These two results are very interesting and worthwhile publication in Nature Communications. Most of experiments have been carried out with reasonable quality and the data by itself are convincing. However, the authors are a bit biased against an idea that the Shu complex recognizes abasic sites during DNA replication and promotes homologous recombination-mediated error-free DNA repair. Possibly, for this reason, they tended over-interpret their results. The authors need to add more data if they would like to claim the idea or just try more fair interpretation of their results with modest wording.

Major points:

1. ChIP to ARS, Figure 1d: The ChIP analysis showed that Csm2 binds to two ARSs under normal growth condition; e.g. asynchronous cultures. This is a bit surprising and new observation. Since recombination proteins are believed to recruit to replication forks when replication forks are stalled. Given the persistent time of a recombination protein on the fork around the ARS seems to be very short, the results suggest that Csm2 binds stably to ARS during cell cycle, like ORC complex. If the Csm2 is a component of the replisome, it might move with ongoing DNA replication, thus hard to detect the binding to ARS, except in early S phase or when DNA replication is stalled. To clarify the results, the authors should do time-course for ChIP analysis in cells starting S phase after alpha-factor arrest at G1. Moreover, it is great if the authors also try the same ChIP time course with perturbed S-phase condition such as HU or MMS treatment.
2. Mutation assay, Figure 2c: The authors measured APOBEC3-induced mutations and found that the *csm2/psy3* mutants showed an increased mutation rate. Interestingly, a mutation in the *UNG1* is epistatic to *csm2/psy3* mutations, suggesting that *UNG1* and *CSM2/PSY3* work in the same pathway to prevent the mutation. However, Figure 2c showed the *csm2/psy3* mutants are different from the *ung1* mutant in mutation spectra. It is interesting to check the mutation spectra in *ung1 csm2* double mutant to know the exact relationship between *UNG1* and Shu genes. Moreover, the authors stressed G-strand bias mutation in *pys3/csm2* mutants (line 148, Figure 2d). However, compared to the *ung1*, G-strand bias is much weaker in the *pys3/csm2* mutants (increase of G-C transversion is clear). The authors could soften the words on the lagging strand bias.
3. AP endonuclease cleavage assay, Figure 4: The authors showed yeast Psy3-Csm2 blocks the cleavage of AP site by AP endonuclease from human. This results sounds very interesting. However, the authors need the control experiments to show specific interaction between Csm2-Psy3 and AP endonuclease. Given Csm2-Psy3 binds to the fork, it is easily predicted that Csm2-Psy3 binding to the fork will block any kind of biochemical reaction at the site. Thus, to deny indirect blocking by Csm2-Psy3 (or to demonstrate unique role of the complex at the AP site in the Y fork), the authors need to use other nuclease such as Mus81-Mms4, or Fen1 endonucleases, which cleave Y-fork efficiently and check the effect of Csm2-Psy3. And also the authors may try Rad51 instead of Csm2-Psy3 for the reaction. Since it is known that Rad51 also binds to this type of branched substrate tightly. So the authors can address whether Rad51 block AP endonuclease activity.
4. In Figure 6c, the higher DNA binding activity of Csm2-Psy3 to substrate "AP6" than "AP7". AP6 and AP7 are almost identical in their DNA structure (except the difference in lengths of ssDNA and dsDNA flanking the AP site). Do the authors have an idea to explain this specific activity of Csm2-

Psy3 to AP6 (not AP7)? It is curious whether the activity of AP endonuclease to AP7 is also affected by Csm2-Psy3 like AP6.

5. All biochemistry in the paper has been done only for Csm2-Psy3 subcomplex. Since Shu1-Shu2 enhances the DNA binding activity of the subcomplex. It is important to analyze the Csm2-Psy3-Shu1-Shu2 tetramer in parallel.

Minor points:

1. Line 91; there is little evidence to support this conclusion that Psy3 stabilizes Shu on DNA since the DNA binding assay uses only measure equilibrium of the association and dissociation to the DNA. Primary conclusion of Csm2-Pys3KRK is that the KRK site of Psy3 somehow contributes the DNA binding activity of Csm2-Pys3. Please soften the wording.

2. Figure 1c: It is very hard to say that the csm2-KRRR psy3-KRK mutant is similar to csm2 null mutant for MMS sensitivity. Thus, simply, csm2-KRRR psy3-KRK mutant shows a partial defect. More surprisingly, csm2-KRRR is quite resistant to MMS (even the authors did not mention). The authors need careful interpretation on these results. It raises the possibility that the DNA binding of Csm2 is not important for Csm2 function in vivo).

3. Extended Figure 2b bottom: there are "two" labels of the csm2-KRRR. One should be for csm2-KRRR psy3-KRK.

4. Line 109: as written in #2, this conclusion is too strong, since the csm2-KRRR psy3KRK mutant is not a null mutant (and also csm2-KRRR mutant is quite proficient)

5. Figure 2a and line 134: Figure 2a shows that the CAN1 gene is located at the left side of ARS. If so, replication fork passes through from the right to the left (thus, leftward). However, line 134, the authors mentioned "rightward". Why discrepancy?

6. Mutation rate measurements, line 274-275: The methods seems to use "median" for the calculation. However, how do the authors get a median for 4 and 6 trials, which are not odd numbers? Moreover, "the averages of each trial were plotted". What "averages" mean in the context? Please clarify my misunderstanding.

Reviewer #2 (Remarks to the Author):

This manuscript describes an investigation of the mechanisms by which Rad51 paralogs (Csm2/Psy3) in budding yeast contribute to genome stability, particularly under conditions of a high burden of abasic sites. This is an important consideration for normal genome maintenance and is a potential therapeutic target for cancer treatment as many cancers are mutated in homologous recombination pathways. Therefore, this paper could be of significant interest to a broader audience. The authors make some novel observations that are generally consistent with the model that they propose, strongly implicating the DNA binding surface of Csm2 as being important for tolerance/repair of abasic DNA sites in yeast. However, it is not clear why it is necessary to invoke specific abasic binding, because the Rad51 paralogs have been recognized in other studies to facilitate Rad51 filament formation and template switching in the context of other types of damage. Overall the quality of the experiments, statistical analysis, and presentation are high, but there are a few experimental issues that could be easily addressed to reach the highest quality possible.

Specific comments:

1. The hypothesis that Csm2-Psy3 specifically recognizes abasic sites is not adequately supported by the data. Some unexplained features of the data are that only 2-fold tighter binding is observed for an abasic analog (tetrahydrofuran) versus a T (Fig 3c). Is this really enough specificity for recognition in vivo? How is the tetrahydrofuran being recognized and what other replication blocks can be bound? Tighter binding is not seen when the tetrahydrofuran is at many other sites in the oligonucleotide. The authors should consider and discuss alternative hypotheses for the interesting cellular effects that they have attributed to this complex. It is not clear how a double-flap structure

could be formed in the cell and this should be better justified.

2. Is tetrahydrofuran an appropriate mimic of an abasic site? If the model is specific recognition of an abasic site, then it may be that a true abasic could bind much tighter and thereby strongly support the proposed model of lesion recognition by Csm2/Psy3. This could be generated from a dU-containing DNA and UNG activity.

3. The language and figures in the manuscript about the leading and lagging strand and comparison of overhang oligonucleotides to a replication fork are confusing and need to be clarified. For example, Fig 3a describes this double-flap structure as both a leading strand and a lagging strand. More explanation is needed for how this structure would be formed at a replication fork.

4. Similarly, Fig 5 is not a unique model to explain the data. It appears that the Shu complex is binding an abasic somewhere between the replicative helicase and the leading strand polymerase in the model. It is not clear what data supports this detail. Given the rapid rate of replication fork movement this would be a very narrow time window for the Shu complex to bind. (See point 5 for an alternative model). If there are multiple models, then these could be discussed in the supplement or a more simplistic model is warranted.

5. Another model that would also be consistent with the data is that Csm2-Psy3 may bind to a 5' overhang (the expected intermediate if a replicative polymerase stalled upon encountering damage and dissociated). An attractive feature of this model is that Csm2-Psy3 could help to recruit Rad51 to stalled replication forks on either the leading or lagging strand and this seems to generally agree with many other papers in the field looking at initiation of homologous recombination and/or template switching by Rad51 paralogs. If literature data on binding to 5' versus 3' overhangs is available, it would be very pertinent to discuss here (to distinguish which model is most likely). If no data is available, then it is important to do a side-by-side comparison using the anisotropy assay. (See review fig 1; A shows how a blocked Okazaki fragment might generate a binding site for Csm2-Psy3; B shows a 5' overhang that could be formed on the lagging strand as in A or by dissociation of the polymerase on the leading strand).

6. As a related point, how does the affinity for a flap or double flap compare to either single-strand or double-strand oligos. This would help to provide context for DNA recognition.

7. What are the "other" category of mutations in Fig 2c,d? Do they provide any insight into the role of Csm2-Psy3?

8. This study relies on fluorescence anisotropy to characterize DNA binding affinity. However, there are some problems with the data in Fig 1b. The theoretical maximum for anisotropy of a stationary fluorophore is 0.4 making it difficult to take the value of 0.65 as being accurate (Supplemental table 1). It is important to report total fluorescence in addition to anisotropy to evaluate if there is an interaction between the protein and the fluorophore (See for example Kozlov et al. (2012), *Methods in Molecular Biology* 922:55-83 and references therein). The Fig 3 legend alludes to sample quality, but it isn't clear what is meant. Why wasn't Fig 1b repeated if there is a problem with it?

Minor comments:

Supplemental Tables 1 and 2 may have more significant figures than warranted

Supplemental Table 5 should state what X is (tetrahydrofuran)

Fig 2c,d – It would be helpful to explain the nomenclature for the mutations (which is the reference strand)

Reviewer #3 (Remarks to the Author):

In the current manuscript, Rosenbaum et al., report that *Saccharomyces cerevisiae* Shu complex proteins promote DNA damage tolerance pathway using an APOBEC3B expression model at the ARS216 chromosome loci that was previously characterized by Roberts group (Hoopes et al., NAR, 2017). The role of Rad51 paralogs including Shu complex proteins in the error free repair of damaged DNA via HR has been known in the literature. Although some of the data presented in the manuscript is interesting, it doesn't provide sufficient novel information and in the current form it is too preliminary for publication in Nature Communications.

1. In the current manuscript authors have NOT examined the canonical yeast paralogs Rad55 and Rad57. Existing literature including work from authors (Godin et al., 2013; Xu et al., 2013; Gaine et al., 2015) show that Shu complex proteins work in conjunction with Rad55-Rad517 Rad51 paralogs. Indeed work from Heyer group shows that Rad55 is essential for HR mediated replication fork recovery (Herzberg et al., 2006). It is surprising why authors have not examined 55-57 complex and a likely interplay between 55-57 with Shu proteins. Without these data, title of the current manuscript is misleading. In fact authors fail to indicate not even once about Shu proteins in the abstract.
2. Authors show that WT-Csm-Psy3 binds to fork structure which is defective with Lysine-Arginine mutations in Fig. 1B. Authors have previously (Gaines et al., 2015) demonstrated that this complex binds proficiently to ssDNA. The data in the Fig. 1B is how relevant? It could be simply binding to ssDNA regions of forked structure. Authors need to carry out these experiments with proper controls.
3. The purified proteins shown in the Extended Fig. 1 should be shown in single gel with Mol. Weight markers.
4. Extended Fig. 2B 4th and 6th bar groups are identical/mislabeled.
5. If Shu proteins are meant for HR mediated repair of replication associated DNA lesions in the whole genome why should it bind only to ARS sites? Rather authors should examine its chromatin association after subjecting it to MMS damage. Authors have not examined specific binding of Shu proteins to fork structures in comparison with ssDNA, dsDNA and other types of replication/recombination intermediate substrates. Author need to measure the affinities of Shu proteins to various substrates and accordingly measure the specificities. If it goes to the sites of damage; how? Is it damage specific independent recruitment or via replication machinery? Authors need to perform more experiments to clarify/validate their observations.
6. In page 6 line 131, authors write "we previously measured... Ref 25, 27..These references are from Roberts group.
7. Increase in the binding affinity of Shu proteins with lagging strand AP site AP6 is not too convincing. Why is that it is only for AP6 and not for AP7 and AP5. Moreover, If Shu proteins have general role in DNA damage tolerance by HR, it should exhibit binding to even leading strand abasic sites. Overall the DNA binding of Shu proteins to forks structure and that too with lagging strand abasic site (only one substrate) is questionable. Authors need to reexamine this with appropriate controls and also by different methods.
8. In fact authors should demonstrate specific *in vivo* binding of Shu complexes to ARS216 loci by ChIP analysis. This should be done again with all the relevant controls and with and without damage.
9. Authors in their Fig.4 by performing *in vitro* experiments show that Shu proteins protect cleavage by AP endonucleases. This is completely an *in vitro* study; any proteins that binds to the fork structures would impede the access of nucleases/other proteins. To prove this point, authors should demonstrate first that Shu proteins indeed localizes to the sites of such AP sites *in vivo* and measure fork collapse/DSB formation. If this was the case, absence of AP endonuclease should rescue *csm2/psy3* sensitivity to MMS but it is opposite with their data presented in Fig.3D.
10. The last paragraph of the discussion about BRCA1/2, PARP inhibitors and treatment of cancer etc.. is irrelevant to the manuscript. First of all there is no report of mammalian Shu complex gene mutations in breast/ovarian cancers.
11. Manuscript is poorly organized.

Response to Reviewer Comments:

Reviewer #1:

1. ChIP to ARS, Figure 1d: The ChIP analysis showed that Csm2 binds to two ARSs under normal growth condition; e.g. asynchronous cultures. This is a bit surprising and new observation. Since recombination proteins are believed to recruit to replication forks when replication forks are stalled. Given the persistent time of a recombination protein on the fork around the ARS seems to be very short, the results suggest that Csm2 binds stably to ARS during cell cycle, like ORC complex. If the Csm2 is a component of the replisome, it might move with ongoing DNA replication, thus hard to detect the binding to ARS, except in early S phase or when DNA replication is stalled. To clarify the results, the authors should do time-course for ChIP analysis in cells starting S phase after alpha-factor arrest at G1. Moreover, it is great if the authors also try the same ChIP time course with perturbed S-phase condition such as HU or MMS treatment.

We agree with the reviewer that the ChIP results demonstrating that Csm2 binds with ARS's under normal growth conditions is surprising and requires further experimentation. We have experienced a number of technical issues with the ChIP experiments doing cell synchronization and in contrast have now had a lot of success with chromatin fractionation experiments, which was a suggestion from Reviewer 3. In this new analysis (**New Figure 2**), we performed chromatin fractionation experiments in the presence or absence of MMS damage after cell cycle arrest and release. To do these experiments, we arrested Csm2-6HA expressing cells in G1 with alpha factor and released these cells into different MMS doses for 1 hour (0%, 0.01%, 0.02%, and 0.03%; **New Figure 2C**). We observe increased Csm2 chromatin association at 0.03% MMS relative to 0% MMS when normalized to the efficiency of extraction. Most strikingly, Csm2-6HA chromatin association increased in a MMS dose-dependent manner (~2-7 fold) when abasic sites accumulate by disrupting *APN1 APN2 NTG1 NTG2*. Lastly, we find that Csm2 chromatin association depends upon its DNA binding residues of Csm2 and Psy3 (**New Figure 2B**). In light of this new analysis and the technical issues we experienced with the ChIPs, we removed the ChIP analysis from the body of the paper and refer to it as an Extended Data Figure in the Discussion and mention that future experimentation is warranted (**Extended Data Figure 9**).

2. Mutation assay, Figure 2c: The authors measured APOBEC3-induced mutations and found that the *csm2/psy3* mutants showed an increased mutation rate. Interestingly, a mutation in the *UNG1* is epistatic to *csm2/psy3* mutations, suggesting that *UNG1* and *CSM2/PSY3* work in the same pathway to prevent the mutation. However, Figure 2c showed the *csm2/psy3* mutants are different from the *ung1* mutant in mutation spectra. It is interesting to check the mutation spectra in *ung1 csm2* double mutant to know the exact relationship between *UNG1* and *Shu* genes. Moreover, the authors stressed G-strand bias mutation in *pys3/csm2* mutants (line 148, Figure 2d). However, compared to the *ung1*, G-strand bias is much weaker in the *pys3/csm2* mutants (increase of G-C transversion is clear). The authors could soften the words on the lagging strand bias.

We appreciate the reviewer's comments regarding this experiment. The APOBEC-induced mutation rates suggest that *UNG1* and *CSM2/PSY3* operate in the same pathway to prevent APOBEC-induced mutations during replication. This interpretation would suggest that *UNG1* operates prior to *CSM2/PSY3* to remove the uracil base formed by APOBEC-induced cytidine deamination and forming a replication fork stalling abasic site. *CSM2/PSY3* would then be

involved in error-free bypass of this abasic site. The reason the *ung1* Δ mutant produces a different spectra from *csm2* or *psy3* mutants is that without *UNG1*, no uracil excision occurs, meaning that uracils generated by APOBEC activity will directly template for G to A mutations, while in the *csm2* Δ /*psy3* Δ mutants, *UNG1* converts the uracils to abasic sites which are then bypassed mutagenically by TLS polymerases resulting in equal numbers of G to A and G to C mutations. We confirmed these roles of *UNG1* and *CSM2* by sequencing *CAN1* mutants in the *ung1* Δ *csm2* Δ double mutant, which produced identical spectra to the *ung1* Δ mutant (i.e. nearly all G to A mutations). This data and explanation have been added to the manuscript in **New Figure 3C and 3D**.

3. AP endonuclease cleavage assay, Figure 4: The authors showed yeast Psy3-Csm2 blocks the cleavage of AP site by AP endonuclease from human. This results sounds very interesting. However, the authors need the control experiments to show specific interaction between Csm2-Psy3 and AP endonuclease. Given Csm2-Psy3 binds to the fork, it is easily predicted that Csm2-Psy3 binding to the fork will block any kind of biochemical reaction at the site. Thus, to deny indirect blocking by Csm2-Psy3 (or to demonstrate unique role of the complex at the AP site in the Y fork), the authors need to use other nuclease such as Mus81-Mms4, or Fen1 endonucleases, which cleave Y-fork efficiently and check the effect of Csm2-Psy3. And also the authors may try Rad51 instead of Csm2-Psy3 for the reaction. Since it is known that Rad51 also binds to this type of branched substrate tightly. So the authors can address whether Rad51 block AP endonuclease activity.

We agree with the reviewers that Csm2-Psy3 binding to a double-flap substrate blocks APE1 endonuclease cleavage and that this is unlikely to be mediated through a direct interaction with APE1. We have now tested this model using *in vitro* pull downs and as expected find that Csm2-Psy3 does not directly interact with APE1 (**New Extended Data Figure 6**).

As the reviewer also suggested here, we have now examined whether Rad51 would also block APE1 cleavage using this same substrate. We find that Rad51 binds to the AP6 substrate with a lower affinity compared to Csm2-Psy3 (**New Extended Data Figure 8**). We next determined whether Rad51 would similarly block APE1 cleavage and indeed it does (**New Figure 5C**). We then examined whether Csm2-Psy3 together with Rad51 would further inhibit APE1 endonuclease activity and we find that APE1 endonuclease activity is further reduced (**New Figure 5D**). Together these results suggest that both Rad51 and Csm2-Psy3 are capable of blocking APE1 endonuclease activity and that Csm2-Psy3 exhibits a higher binding affinity for the AP6 compared to Rad51.

Since we prioritized the known interacting partners of Csm2-Psy3 for our analysis, Rad51 or Shu1-Shu2 (see below), we did not test other nucleases such as Mus81-Mms4 or Fen1.

4. In Figure 6c, the higher DNA binding activity of Csm2-Psy3 to substrate “AP6” than “AP7”. AP6 and AP7 are almost identical in their DNA structure (except the difference in lengths of ssDNA and dsDNA flanking the AP site). Do the authors have an idea to explain this specific activity of Csm2-Psy3 to AP6 (not AP7)? It is curious whether the activity of AP endonuclease to AP7 is also affected by Csm2-Psy3 like AP6.

This is truly a remarkable result and we are trying to get co-crystals with this substrate and Csm2-Psy3 to address the question raised by the reviewer. It is possible that there is a binding pocket in the Csm2-Psy3 complex that can accommodate an abasic site when it is only in the

AP6 position and not when it is in the other positions. This binding pocket allows increased binding affinity which we have clearly observed. We have included this possibility in the Results.

5. All biochemistry in the paper has been done only for Csm2-Psy3 subcomplex. Since Shu1-Shu2 enhances the DNA binding activity of the subcomplex. It is important to analyze the Csm2-Psy3-Shu1-Shu2 tetramer in parallel.

Although we are not aware of study where Shu1-Shu2 enhances DNA binding activity of the Csm2-Psy3 subcomplex, we agree with the reviewer that it would be insightful to include Shu1-Shu2 in our studies. We initially focused on Csm2-Psy3 alone because Csm2-Psy3 are the DNA binding subunits, whereas Shu1-Shu2 do not bind DNA at concentrations up to 800 nM, which we also observed by EMSA (**New Extended Data Figure 5**). In response to the reviewer comments, we have now repeated the key findings in the presence Shu1-MBP-Shu2 provided by our collaborator, Patrick Sung. We do not observe Shu1-MBP-Shu2 binding to AP6 (**New Extended Data Figure 5a**) nor do we find that Shu1-MBP-Shu2 stimulates Csm2-Psy3 binding to AP6 (**New Figure 4D and New Extended Data Figure 5b**). Furthermore, the addition of Shu1-MBP-Shu2 to Csm2-Psy3 does not significantly protect the AP6 fork from APE1 cleavage (**New Figure 5E**). Therefore, we find that Csm2-Psy3 DNA binding activities are Shu1-Shu2 independent.

Minor points:

1. Line 91; there is little evidence to support this conclusion that Psy3 stabilizes Shu on DNA since the DNA binding assay uses only measure equilibrium of the association and dissociation to the DNA. Primary conclusion of Csm2-Pys3KRK is that the KRK site of Psy3 somehow contributes the DNA binding activity of Csm2-Pys3. Please soften the wording.

We have now softened the language as suggested.

2. Figure 1c: It is very hard to say that the csm2-KRRR psy3-KRK mutant is similar to csm2 null mutant for MMS sensitivity. Thus, simply, csm2-KRRR psy3-KRK mutant shows a partial defect. More surprisingly, csm2-KRRR is quite resistant to MMS (even the authors did not mention). The authors need careful interpretation on these results. It raises the possibility that the DNA binding of Csm2 is not important for Csm2 function in vivo).

We agree with the reviewer and have softened the language. We think that our results suggest that *in vivo* both Csm2 and Psy3 DNA binding activities contribute to Shu complex function in resistance to MMS-induced DNA damage.

3. Extended Figure 2b bottom: there are “two” labels of the csm2-KRRR. One should be for csm2-KRRR psy3-KRK.

Corrected.

4. Line 109: as written in #2, this conclusion is too strong, since the csm2-KRRR psy3KRK mutant is not a null mutant (and also csm2-KRRR mutant is quite proficient)

We appreciate the reviewer’s comment and have now softened the language.

5. Figure 2a and line 134: Figure 2a shows that the CAN1 gene is located at the left side

of ARS. If so, replication fork passes through from the right to the left (thus, leftward). However, line 134, the authors mentioned “rightward”. Why discrepancy?

We thank the reviewer for catching this discrepancy. The fork does in fact move leftward through *CAN1*. The discrepancy has been corrected.

6. Mutation rate measurements, line 274-275: The methods seems to use “median” for the calculation. However, how do the authors get a median for 4 and 6 trials, which are not odd numbers? Moreover, “the averages of each trial were plotted”. What “averages” mean in the context? Please clarify my misunderstanding.

We thank the reviewer for pointing out this. We used FALCOR (Fluctuation AnaLysis CalculatOR) with the Lea-Coulson Method of the Median (LC) to calculate mutation rates. For each trial, 5 individual clones per genotype were analyzed and a median mutation rate determined. The experiment was repeated 4 to 6 times depending on the strain analyzed and then the individual media mutation rates from those experiments were averaged. We have now clarified this in the Materials and Methods.

Reviewer #2:

1. The hypothesis that Csm2-Psy3 specifically recognizes abasic sites is not adequately supported by the data. Some unexplained features of the data are that only 2-fold tighter binding is observed for an abasic analog (tetrahydrofuran) versus a T (Fig 3c). Is this really enough specificity for recognition *in vivo*? How is the tetrahydrofuran being recognized and what other replication blocks can be bound? Tighter binding is not seen when the tetrahydrofuran is at many other sites in the oligonucleotide. The authors should consider and discuss alternative hypotheses for the interesting cellular effects that they have attributed to this complex. It is not clear how a double-flap structure could be formed in the cell and this should be better justified.

We thank the reviewer for these insightful comments. We have provided several lines of evidence to support the model that the Shu complex is important for the error-free tolerance of abasic sites both *in vitro* and *in vivo*. This conclusion is now further substantiated by additional experimentation: First, in the original manuscript, we showed that *csm2Δ* or *psy3Δ* cells promote bypass of DNA replication-associated APOBEC3B-induced lesions, which are converted to abasic sites by the Ung1 DNA glycosylase (**New Figure 3**) and that DNA binding residues of Csm2 and Psy3 are important for MMS resistance when abasic sites accumulate by disruption of *apn1Δ apn2Δ ntg1Δ ntg2Δ* (**New Figure 2A**). Secondly, *in vitro* data show that Csm2-Psy3 has a higher affinity for DNA double-flap structure with an abasic site analog on the ssDNA region proximal to the flap (AP6; **New Figure 4C**). Third, we also now show that Csm2 chromatin association is significantly enriched when abasic sites accumulate upon MMS exposure in a dose dependent manner and that this enrichment depends on its DNA binding residues (**New Figure 2**). Together this *in vitro* and *in vivo* evidence strengthens the notion that the Shu complex is important for error-free tolerance of abasic sites through its DNA binding activities.

We agree with the reviewer that abasic sites may not be the only lesion that the Shu complex recognizes and we have included this possibility in the Discussion.

We agree with the reviewer that the *in vitro* double-flap structure we used is not formed in a cell. We have addressed this point in detail in Reviewer comment #3.

2. Is tetrahydrofuran an appropriate mimic of an abasic site? If the model is specific recognition of an abasic site, then it may be that a true abasic could bind much tighter and thereby strongly support the proposed model of lesion recognition by Csm2/Psy3. This could be generated from a dU-containing DNA and UNG activity.

Tetrahydrofuran (THF) only differs from an abasic site by a H on the 1' position, which prevents the open sugar ring aldehyde to form, and thus prevents spontaneous beta-elimination and strand breaking. Since THF is more stable than a true abasic site it has been extensively used to mimic an abasic site in a wide range of biochemical studies. Tomas Lindahl, Mark Kelley, Samuel Wilson, and David Wilson.

To address this concern, we created double-flap structure that would be a substrate for Ung1 with a dU at the same position as the THF in AP6. We then added Ung1 to remove the dU and create an abasic site. Unfortunately, the background cleavage was unacceptable for our analysis. Therefore, we are unable to use a real abasic site for our analysis.

3. The language and figures in the manuscript about the leading and lagging strand and comparison of overhang oligonucleotides to a replication fork are confusing and need to be clarified. For example, Fig 3a describes this double-flap structure as both a leading strand and a lagging strand. More explanation is needed for how this structure would be formed at a replication fork.

We appreciate this concern, and we apologize that we did not accurately describe the oligonucleotide substrates that were used and have changed the nomenclature accordingly throughout. In addition, to address the point of how a double-flap structure would be formed at a replication fork, we have now analyzed Csm2-Psy3 DNA binding to an abasic site analog along the ssDNA region proximal to a 5'-flap, 3'-flap, a DNA structure that more closely resembles a fork with two dsDNA regions (**New Extended Data Figure 4**). We find that Csm2-Psy3 bind to all of these substrates with similar binding affinities by EMSA.

4. Similarly, Fig 5 is not a unique model to explain the data. It appears that the Shu complex is binding an abasic somewhere between the replicative helicase and the leading strand polymerase in the model. It is not clear what data supports this detail. Given the rapid rate of replication fork movement this would be a very narrow time window for the Shu complex to bind. (See point 5 for an alternative model). If there are multiple models, then these could be discussed in the supplement or a more simplistic model is warranted.

We agree with the reviewer that there may be multiple models that could fit our data. The reviewer brings up some specific models in the subsequent point that we discuss below #5 and have incorporated into the manuscript.

5. Another model that would also be consistent with the data is that Csm2-Psy3 may bind to a 5' overhang (the expected intermediate if a replicative polymerase stalled upon encountering damage and dissociated). An attractive feature of this model is that Csm2-Psy3 could help to recruit Rad51 to stalled replication forks on either the leading or

lagging strand and this seems to generally agree with many other papers in the field looking at initiation of homologous recombination and/or template switching by Rad51 paralogs. If literature data on binding to 5' versus 3' overhangs is available, it would be very pertinent to discuss here (to distinguish which model is most likely). If no data is available, then it is important to do a side-by-side comparison using the anisotropy assay. (See review fig 1; A shows how a blocked Okazaki fragment might generate a binding site for Csm2-Psy3; B shows a 5' overhang that could be formed on the lagging strand as in A or by dissociation of the polymerase on the leading strand).

In a previous study [Godin et al (2013) *Nucleic Acids Research* 41(8): 4525-2534], we directly compared the binding affinity of Csm2-Psy3 for double-flap DNA by competing with unlabeled substrates (3' overhang, 5' overhang, dsDNA, ssDNA, and an unlabeled double-flap; **Figure 1B** from Godin et al). We determined that the unlabeled double-flap DNA is the best competitor (apparent $K_i = 98$ nM) followed by a 3' and 5' overhang (apparent $K_i = 188$ nM and 224 nM, respectively). We have now discussed this data and an alternative model in the Discussion.

6. As a related point, how does the affinity for a flap or double flap compare to either single-strand or double-strand oligos. This would help to provide context for DNA recognition.

As mentioned in point #5, in a previous manuscript by Godin et al (2013) *NAR* 41(8): 4525, we directly compared the binding affinity of Csm2-Psy3 for double-flap DNA by competing with unlabeled substrates such as dsDNA or ssDNA. The dsDNA and ssDNA were the worst competitors exhibiting an apparent $K_i =$ of 334 nM and 3190 nM, respectively.

7. What are the “other” category of mutations in Fig 2c,d? Do they provide any insight into the role of Csm2-Psy3?

The “other” category of mutations consists primarily of short insertions and deletions and complex mutation events (i.e. usually a base substitution with a neighboring insertion or deletion). A very small number of substitutions at A and T bases are also included in this category because they are unlikely to be induced by the APOBEC3B and are not needed for our analyses. Since the “Other” category mutants only constitute ~10% of all mutations (only 37 events total) assessed, making any interpretation of their mechanism is difficult. We do believe that the complex events in the csm2-psy3 mutants may stem from forcing bypass of APOBEC3B-induced abasic sites by TLS polymerases. These polymerases are highly error-prone and could be responsible for the base substitution at the abasic site and then they could induce a secondary frameshift mutation. However, verification of this mechanism will require significantly more work that is beyond the scope of this manuscript. We are currently pursuing this as another project. We have specified what mutations are included in the other category in the text.

8. This study relies on fluorescence anisotropy to characterize DNA binding affinity. However, there are some problems with the data in Fig 1b. The theoretical maximum for anisotropy of a stationary fluorophore is 0.4 making it difficult to take the value of 0.65 as being accurate (Supplemental table 1). It is important to report total fluorescence in addition to anisotropy to evaluate if there is an interaction between the protein and the fluorophore (See for example Kozlov et al. (2012), *Methods in Molecular Biology* 922:55-83 and references therein). The Fig 3 legend alludes to sample quality, but it isn't clear

what is meant. Why wasn't Fig 1b repeated if there is a problem with it?

We apologize for this mistake and it has now been corrected. We didn't set an end protein concentration value when fitting the data and so the Bmax was extrapolated.

In addition, the sample quality issue between **Figure 1B** and **New Figure 4** is due to mutant Csm2-Psy3 dimers failing to bind the heparin column and therefore instead these proteins were all purified using a HiTrap Q (GE Healthcare) anion exchange column. To avoid bias, we also purified the wild-type Csm2-Psy3 dimers used in the same manner so that we could make unbiased comparisons. This is detailed in the Materials and Methods.

Minor comments:

1. Supplemental Tables 1 and 2 may have more significant figures than warranted

Corrected

2. Supplemental Table 5 should state what X is (tetrahydrofuran)

Corrected

3. Fig 2c,d – It would be helpful to explain the nomenclature for the mutations (which is the reference strand)

The reference strand refers to the Watson DNA strand.

Reviewer #3:

1. In the current manuscript authors have NOT examined the canonical yeast paralogs Rad55 and Rad57. Existing literature including work from authors (Godin et al., 2013; Xu et al., 2013; Gaine et al., 2015) show that Shu complex proteins work in conjunction with Rad55-Rad517 Rad51 paralogs. Indeed work from Heyer group shows that Rad55 is essential for HR mediated replication fork recovery (Herzberg et al., 2006). It is surprising why authors have not examined 55-57 complex and a likely interplay between 55-57 with Shu proteins. Without these data, title of the current manuscript is misleading. In fact authors fail to indicate not even once about Shu proteins in the abstract.

We agree with the reviewer that it would have been ideal to include Rad55-Rad57 in our analysis. Our previous work from Gaines et al (2015) represented, to our knowledge, the largest eukaryotic *in vitro* reconstitution of Rad51 filament assembly and this analysis was performed in Dr. Patrick Sung's laboratory. Rad55-Rad57 are technically challenging to purify and therefore only a few laboratories (such as Heyer and Sung) have been able to do so. However, we have now included data that includes Rad51 and Shu1-Shu2 for the key assays described. We have also now specified the Shu complex in the abstract.

2. Authors show that WT-Csm-Psy3 binds to fork structure which is defective with Lysine-Arginine mutations in Fig. 1B. Authors have previously (Gaines et al., 2015) demonstrated that this complex binds proficiently to ssDNA. The data in the Fig. 1B is how relevant? It could be simply binding to ssDNA regions of forked structure. Authors need to carry out these experiments with proper controls.

As mentioned in Reviewer 2 point #5 and #6, in a previous manuscript by Godin et al (2013) NAR 41(8): 4525, we directly compared the binding affinity of Csm2-Psy3 for double-flap DNA by competing with unlabeled substrates such as dsDNA or ssDNA. The dsDNA and ssDNA were the worst competitors exhibiting an apparent $K_i = 334$ nM and 3190 nM, respectively. So although the Shu complex does bind to ssDNA, its binding affinity is greatly reduced compared to a double-flap substrate. We respectfully disagree with the reviewer that we have not carried out the proper controls because these controls were performed and described in our previous paper. We have now discussed these results in the manuscript.

3. The purified proteins shown in the Extended Fig. 1 should be shown in single gel with Mol. Weight markers.

We have now run all the purified proteins used in this study on a single gel with molecular weight markers, **New Extended Data Figure 1**.

4. Extended Fig. 2B 4th and 6th bar groups are identical/mislabeled.

Corrected.

5. If Shu proteins are meant for HR mediated repair of replication associated DNA lesions in the whole genome why should it bind only to ARS sites? Rather authors should examine its chromatin association after subjecting it to MMS damage.

We thank the reviewer for this helpful suggestion and have now performed chromatin fractionation experiments in the presence or absence of MMS damage after cell cycle arrest and release. To do these experiments, we arrested Csm2-6HA expressing cells in G1 with alpha factor and released these cells into different MMS doses for 1 hour (0%, 0.01%, 0.02%, and 0.03%; **New Figure 2C**). We observe increased Csm2 chromatin association at 0.03% MMS relative to 0% MMS and when normalized to the efficiency of extraction. Most strikingly, Csm2-6HA chromatin association increased in an MMS dose-dependent manner (~2-7 fold) when abasic sites accumulate by disrupting *APN1 APN2 NTG1 NTG2*. Lastly, we find that Csm2 chromatin association depends upon its DNA binding residues of Csm2 and Psy3 (**New Figure 2B**).

Authors have not examined specific binding of Shu proteins to fork structures in comparison with ssDNA, dsDNA and other types of replication/recombination intermediate substrates. Author need to measure the affinities of Shu proteins to various substrates and accordingly measure the specificities.

We have addressed this point in Reviewer Comment #2.

If it goes to the sites of damage; how? Is it damage specific independent recruitment or via replication machinery? Authors need to perform more experiments to clarify/validate their observations.

We have shown that Csm2-Psy3 directly bind to double-flapped DNA substrates *in vitro* and that *in vivo* the DNA binding residues are critical for chromatin immunoprecipitation of Csm2 to multiple ARS's but not to intergenic regions. As described above, we have now performed chromatin fraction experiments and again observe that Csm2 chromatin association depends on its DNA binding residues (**New Figure 2B**), and have thus validated the results. It is possible

that the replication machinery or other HR factors contribute to Shu complex recruitment and we have now included this possibility in the Discussion.

6. In page 6 line 131, authors write “we previously measured... Ref 25, 27..These references are from Roberts group.

Corrected.

7. Increase in the binding affinity of Shu proteins with lagging strand AP site AP6 is not too convincing. Why is that it is only for AP6 and not for AP7 and AP5. Moreover, If Shu proteins have general role in DNA damage tolerance by HR, it should exhibit binding to even leading strand abasic sites. Overall the DNA binding of Shu proteins to forks structure and that too with lagging strand abasic site (only one substrate) is questionable. Authors need to reexamine this with appropriate controls and also by different methods.

This is truly a remarkable result and we are trying to get co-crystals with this substrate and Csm2-Psy3 to address the question raised by the reviewer. It is possible that there is a binding pocket in the Csm2-Psy3 complex that can accommodate an abasic site when it is only in the AP6 position and not when it is in the other positions. This binding pocket allows increased binding affinity which we have clearly observed. We have included this possibility in the Results.

8. In fact authors should demonstrate specific in vivo binding of Shu complexes to ARS216 loci by ChIP analysis. This should be done again with all the relevant controls and with and without damage.

We have now performed ChIP analysis of Csm2 recruitment to ARS216 and indeed observe enrichment (**New Extended Data Figure 9**). In addition, as discussed in Comment #5, we have now performed chromatin fractionation experiments analyzing Csm2 chromatin association in synchronized cells upon increasing MMS doses (**New Figure 2C**), when abasic sites accumulate (**New Figure 2B**), and when the DNA binding residues of Csm2 and Psy3 are mutated (**New Figure 2B**).

9. Authors in their Fig.4 by performing in vitro experiments show that Shu proteins protect cleavage by AP endonucleases. This is completely an in vitro study; any proteins that binds to the fork structures would impede the access of nucleases/other proteins. To prove this point, authors should demonstrate first that Shu proteins indeed localizes to the sites of such AP sites in vivo and measure fork collapse/DSB formation. If this was the case, absence of AP endonuclease should rescue csm2/psy3 sensitivity to MMS but it is opposite with their data presented in Fig.3D.

Unfortunately, it would be technically challenging to demonstrate that the Shu complex proteins localize to AP sites *in vivo*. Unlike an endonuclease-induced DSB, AP sites cannot be induced at a specific genomic locus and endogenous AP sites are not stable. The best we can do to address this comment is to show that Csm2 enrichment at chromatin is enhanced when AP sites accumulate in a DNA binding dependent manner (**New Figure 2B**).

10. The last paragraph of the discussion about BRCA1/2, PARP inhibitors and treatment of cancer etc.. is irrelevant to the manuscript. First of all there is no report of mammalian Shu complex gene mutations in breast/ovarian cancers.

We respectfully disagree with this comment. We believe that the BRCA1/2 and PARP discussion is important because it puts our work into a broader context. Three of the four yeast Shu complex members are Rad51 paralogs and mutations in the human Rad51 paralogs are associated with breast and ovarian cancer predisposition. Furthermore, several RAD51 paralogs are now included in the breast and ovarian cancer screening panels. In addition, we participated in a study that examined patients with mutations in RAD51 paralogs, RAD51C and RAD51D, for PARP sensitivity and resistance mechanisms [Kondrashova et al (2017) Cancer Discovery 7(9): 984]. Unfortunately, we do not know which yeast Rad51 paralogs correspond to their mammalian counterparts. Therefore, we believe that our discussion of the human RAD51 paralogs is important because it could potentially indicate new drug targets, mechanism of action, and context for future studies in human cells.

11. Manuscript is poorly organized.

The extensive editorial revisions and additional data included should now make the manuscript more organized.

Reviewers' comments:

Reviewer #1 (Remarks to the Author):

NCOMMS-18-14750A

Rosenbaum et al.

The revised version of the paper by Rosenbaum et al. describe the role of the yeast Shu complex, whose major subunits contains Psy3 and Csm2, Rad51 paralogs, in the repair of stalled replication forks at abasic (apurinic and apyrimidic; AP) sites. The authors identified critical residues for DNA binding activity of Csm2-Psy3 dimers. Purified Csm2-Pys3 inhibits an activity of human AP endonuclease on Y-fork DNA substrates containing an abasic site at the junction. This conclusion is supported by genetic assays. The authors added several in vivo and in vitro assays in responses to the reviewers. These results are very interesting and worthwhile publication in Nature Communications. Some experiments were added and, with rewriting, the current version is better than in the previous one to support the original idea. Although the authors added more data for the revision than the previous one, it needs some works to explain their interesting, but "strange", results by adding the data and/or more writing.

Major points:

1. The authors added a new result of chromatin fractionation assay (Figure 2), which shows that Csm2 binds to chromatin when DNA is damaged. This is very interesting by itself. However, this is very inconsistent with their previous ChIP (Moved to Extended Figure 9) results that Csm2 binds to ARS under normal condition. If the protein was recovered in chromatin under normal condition and its binding is increased under damage condition, it would be consistent with the ChIP. How do the authors explain this discrepancy?
2. In my previous review, I asked the authors to explain the difference of binding affinity between substrates "AP6" and "AP7" (Figure 4). Unfortunately, because of my writing, the authors misunderstood what I suggested. AP6 has T/AP (abasic site) while AP7 does G/AP. This suggests that the recognition of the AP sites is sequence-context dependent. My suggestion is to do the binding assay using AP6 derivatives with G/AP (or C/AP, A/AP) and AP7 derivative with T/AP etc. As pointed out by other reviewers, the double-flap DNAs is not a natural in vivo substrate. It needs more careful examination for the binding specificity. In the same line, please add the analysis of AP7 in the assay using Psy3-Csm2-Shu1-Shu2 tetramer in Figure 4d.
3. In the same as in #2 point, it is very nice for the authors to add the different assay such as EMSA in extended Figure 4 and 5, which strengthen the conclusion. However, it would be nice to show a raw data of gels in the Figure e4. And should be included the analysis using AP7 as a substrate with AP6.
4. Fig 1b (and line 127) shows that Kd of Csm2-Psy3 to the double flap is 435 +/- 37 nM. However, a previous report by the same group (Gordin et al. 2013) indicated the Kd to the same substrate is 361 +/- 11 nM. On the other hand, in Fig. 4b, the value is 127 +/- 10. This suggests that the lengths and/or sequences of substrates DNAs, rather than the fork per se (or assay conditions), are important determinant for the binding of the Csm2-Psy3 dimer. Why the values are different? This is very important in order to evaluate the key results in Figure 4. I am sorry that I did not find this difference in my previous review.
5. As pointed out by one of the reviewer, the relationship of the activity of Csm3-Psy3 which the authors described in this paper to Rad55-Rad57 sounds very important based on the authors' paper (Gaines et al., Nature Communications, 2015). Of course the biochemistry with Rad55-57 might be a next step. At least the authors carried out sensitivity and mutation assays in Fig. 1, 2 and 3 for csm2 rad55 ung1 etc.

Minor points (sorry that I did not mention these in previous my review):

1. Fig 1d; a mutation rate in psy3-KRK is very similar to that in wild type, but the authors say the difference are significant. I do not think that student's t-test is not applicable to these comparisons (if so, please show the values follows normal distribution). It would be better to use "dynamite"

plots in the all Figures; please use scattered plots with bar blots. Moreover, what do error bars indicate? Please provide it in the legend.

2. Figure 5: Please write the order of the addition of APE1 and Csm2-Psy3.

3. Figure 5c: Please quantify the result.

Reviewer #2 (Remarks to the Author):

I find that the authors have greatly improved the rigor and overall clarity of the manuscript and that this is an interesting paper that provides novel mechanistic insights into how DNA damage is recognized during DNA replication.

I have a very minor suggestion regarding the new extended figure 4. This experiment does not indicate the number of replicates. Although the legend and methods says protein complex, it looks like there are at least 2 complexes observed in extended data figure 5. Possibly referring to extended data figure 5 as representative data for this experiment will help the interested reader to understand the experiment. This could be fixed at the proof stage.

Reviewer #3 (Remarks to the Author):

Authors have partially addressed my comments but the major concerns regarding the DSG repair by Rad51 paralogs remain unanswered.

1. Authors have analyzed the chromatin binding of WT- and KRRR-Csm2 after MMS treatment and claim that while WT protein is loaded onto the chromatin, mutant protein fails to load. I have two issue with the result presented.

A. Authors should include a non-template inducing replication stress (such as HU, Aphidicolin) and analyze the chromatin loading of Csm2. This experiment is essential to claim the specific role of shu complex during lesion bypass.

B. I am not convinced that KRRR mutation abrogates chromatin loading of Csm2. The level of decrease (Fig. 2b) in chromatin association of csm2 variant is not sufficient to make a strong claim that chromatin enrichment depends on Csm2 DNA binding. Authors fail to include any positive controls (such as PCNA or RPA) in this important experiment.

2. The major claim of the manuscript is Shu proteins participating in DSG repair by HR. Authors by in vitro assays show that Csm2/Psy3 prevents DSB formation at fork sites by AP endonuclease. This is not supported by in vivo data. Authors have not shown physical generation of DSBs (by PFGE or COMET assay) in the absence of csm2/psy3 after treatment with various replication stress. Moreover, authors fail to demonstrate that AP endonuclease mutation can rescue csm2/psy3 mutant cells from MMS induced genotoxic stress. Rather they see an opposite phenotype (Fig. 2a). This is a conflicting data with their in vitro study. Without conclusive data how can authors claim and state that this is the first evidence for the role of Shu proteins in damage tolerance.

Response to Reviewers' comments:

Reviewer #1:

1. The authors added a new result of chromatin fractionation assay (Figure 2), which shows that Csm2 binds to chromatin when DNA is damaged. This is very interesting by itself. However, this is very inconsistent with their previous ChIP (Moved to Extended Figure 9) results that Csm2 binds to ARS under normal condition. If the protein was recovered in chromatin under normal condition and its binding is increased under damage condition, it would be consistent with the ChIP. How do the authors explain this discrepancy?

We understand the reviewers point about the difference between the chromatin fractionation experiments and the ChIP result. We think this difference is due to the nature of the experiments and the detection methods. Whereas the chromatin fractionation analyses bulk chromatin enrichment, the ChIP uses a PCR-based strategy for specific loci, which greatly amplifies the signal in a way that may not be readily detectable in bulk assays. It is possible that Csm2 is chromatin associated without damage and we find that its association is further enriched upon accumulation of abasic sites. To clarify this point, we have changed the language in the results to stress that the chromatin association is “enriched” upon abasic site accumulation and not absent under normal conditions. Note that our collaborators updated the ChIP analysis and these changes were incorporated into the graph and do not alter the conclusions.

2. In my previous review, I asked the authors to explain the difference of binding affinity between substrates “AP6” and “AP7” (Figure 4). Unfortunately, because of my writing, the authors misunderstood what I suggested. AP6 has T/AP (abasic site) while AP7 does G/AP. This suggests that the recognition of the AP sites is sequence-context dependent. My suggestion is to do the binding assay using AP6 derivatives with G/AP (or C/AP, A/AP) and AP7 derivative with T/AP etc. As pointed out by other reviewers, the double-flap DNAs is not a natural in vivo substrate. It needs more careful examination for the binding specificity. In the same line, please add the analysis of AP7 in the assay using Psy3-Csm2-Shu1-Shu2 tetramer in Figure 4d.

We now understand the reviewer’s concern more clearly and have repeated the Csm2-Psy3 DNA binding assays using AP6 and AP7 substrates with different nucleotides across from the THF site. Importantly, we do not observe changes in DNA binding affinity when other nucleotides are used (New Extended Data Figure 5a).

3. In the same as in #2 point, it is very nice for the authors to add the different assay such as EMSA in extended Figure 4 and 5, which strengthen the conclusion. However, it would be nice to show a raw data of gels in the Figure e4. And should be included the analysis using AP7 as a substrate with AP6.

We have now included gels and added additional replicates with multiple protein preparations because we observed decreased binding with the static replication fork (n=4). These results are consistent with reduced dsDNA binding of Csm2-Psy3 (Godin et al 2013) and we have now modified the text.

4. Fig 1b (and line 127) shows that Kd of Csm2-Psy3 to the double flap is 435±37 nM. However, a previous report by the same group (Gordin et al. 2013) indicated the Kd to the same substrate is 361±11 nM. On the other hand, in Fig. 4b, the value is 127±10. This suggests that the lengths and/or sequences of substrates DNAs, rather than the fork per se (or assay conditions), are important determinant for the binding of the Csm2-Psy3 dimer. Why the values are different? This is very important in order to evaluate the key results in Figure 4. I am sorry that I did not find this difference in my previous review.

We agree with the reviewer and understand the point of confusion here. The different binding affinity between **Figure 1b** and **4b** is due to the mutant Csm2-Psy3 dimers failing to bind the heparin column and therefore instead these proteins were all purified using a HiTrap Q (GE Healthcare) anion exchange column. To avoid bias, we also purified the wild-type Csm2-Psy3 dimers in the same manner so that we could make direct comparisons between the wild-type and DNA binding mutant Csm2-Psy3 proteins. This is detailed in the Materials and Methods and also mentioned in the Figure legend. Lastly, the change in the binding affinity from the Godin et al 2013 paper could be explained by refining of our binding model based upon new mechanistic insight. The previous publication fit the data to a Hill equation (Godin et al) and here we used a hyperbolic one site binding equation.

5. As pointed out by one of the reviewer, the relationship of the activity of Csm3-Psy3 which the authors described in this paper to Rad55-Rad57 sounds very important based on the authors' paper (Gaines et al., Nature Communications, 2015). Of course the biochemistry with Rad55-57 might be a next step. At least the authors carried out sensitivity and mutation assays in Fig. 1, 2 and 3 for csm2 rad55 ung1 etc.

We agree with the reviewer that analysis using Rad55-Rad57 is a logical next step and is beyond the scope of the current manuscript.

Minor points (sorry that I did not mention these in previous my review):

1. Fig 1d; a mutation rate in psy3-KRK is very similar to that in wild type, but the authors say the difference are significant. I do not think that student's t-test is not applicable to these comparisons (if so, please show the values follows normal distribution). It would be better to use "dynamite" plots in the all Figures; please use scattered plots with bar blots. Moreover, what do error bars indicate? Please provide it in the legend.

We agree with the reviewer and have overlaid a scatter plot with the bar graphs for the chromatin enrichment experiments. The error bars are standard deviations and are indicated in the legend.

For the mutagenesis assays with the DNA binding mutants, we reanalyzed the data from Figure 1D in the same manner described for Figure 3C and do not observe any differences with the previous analyses. For this analysis, all the experimental trials (five independent colonies for 4-6 experiments) were used to find a single mutation rate value and this prevents us from plotting a scatter plot. However, we now show 95% confidence intervals obtained from the FALCOR fluctuation analysis and therefore we removed the t-test comparison.

2. Figure 5: Please write the order of the addition of APE1 and Csm2-Psy3.

Done.

3. Figure 5c: Please quantify the result.

Done.

Reviewer #2:

I have a very minor suggestion regarding the new extended figure 4. This experiment does not indicate the number of replicates.

Corrected (n=4).

Although the legend and methods says protein complex, it looks like there are at least 2 complexes observed in extended data figure 5. Possibly referring to extended data figure 5 as representative data for this experiment will help the interested reader to understand the experiment. This could be fixed at the proof stage.

Because the protein complex is most stable with high glycerol, these EMSA reactions were run in high glycerol gels with a mix of NP40 and dI/dC (mixture of 200 bp dsDNA) to reduce non-specific dsDNA binding between protein and dye-labeled DNA and this can change the mobility of the complexes. Unfortunately, because this is a non-equilibrium assay, we cannot conclude if there are multiple complexes. We have added this point to the figure legend.

Reviewer #3:

1. Authors have analyzed the chromatin binding of WT- and KRRR-Csm2 after MMS treatment and claim that while WT protein is loaded onto the chromatin, mutant protein fails to load. I have two issue with the result presented.

A. Authors should include a non-template inducing replication stress (such as HU, Aphidicolin) and analyze the chromatin loading of Csm2. This experiment is essential to claim the specific role of shu complex during lesion bypass.

We agree with the reviewer and how now included analysis with 50 mM and 200 mM HU where we do not observe Csm2 chromatin enrichment in WT cells or when abasic sites accumulate from *apn1Δ apn2Δ ntg1Δ ntg2Δ* relative to untreated (**New Extended Data Figure 4b**).

B. I am not convinced that KRRR mutation abrogates chromatin loading of Csm2. The level of decrease (Fig. 2b) in chromatin association of csm2 variant is not sufficient to make a strong claim that chromatin enrichment depends on Csm2 DNA binding. Authors fail to include any positive controls (such as PCNA or RPA) in this important experiment.

We agree with the reviewer and now include RPA as a positive control. We repeated the experiment showing that Csm2-KRRR DNA binding mutant does not become chromatin enriched upon MMS damage but that Rfa1 (large subunit of RPA) is chromatin associated (New Extended Data Figure 4a).

2. The major claim of the manuscript is Shu proteins participating in DSG repair by HR.

Figure 4. *CSM2* is important for repair of MMS-induced damage during S phase but not G2 phase. (A) Protein levels of Csm2, Clb2 (S/G2-phase control) and Kar2 (loading control) in asynchronous (ASN) cells or after release from G1 arrest by alpha factor. (B) Chromosome shattering and restitution in WT, *apn1Δ apn2Δ* and *apn1Δ apn2Δ csm2Δ* cells arrested in G2 cell cycle by nocodazole. Chromosomes are harvested at the indicated time points before and after treatment with MMS and separated by PFGE as described in the methods. (C) Chromosome restitution following progression through S phase in WT or *csm2Δ* cells treated with MMS.

Authors by *in vitro* assays show that Csm2/Psy3 prevents DSB formation at fork sites by AP endonuclease. This is not supported by *in vivo* data. Authors have not shown physical generation of DSBs (by PFGE or COMET assay) in the absence of *csm2/psy3* after treatment with various replication stress. Moreover, authors fail to demonstrate that AP endonuclease mutation can rescue *csm2/psy3* mutant cells from MMS induced genotoxic stress. Rather they see an opposite phenotype (Fig. 2a). This is a conflicting data with their *in vitro* study. Without conclusive data how can authors claim and state that this is the first evidence for the role of Shu proteins in damage tolerance.

We respectfully disagree that our *in vitro* data is not supported by the *in vivo* work. Previous findings from the Rothstein group [Shor et al (2005) Genetics 169(3): 1275] has demonstrated that Rad52 foci, likely indicative of DSBs, accumulate and persist in Shu complex mutants upon MMS damage. Secondly, in a previous manuscript [Godin et al (2016) NAR 30(44): 8199], we showed by PFGE that *csm2Δ* cells exhibit a delay in reconstitution of chromosomes

following MMS damage (Published Figure 4C on left). Furthermore, we did not observe an additional delay or accumulation of DSBs when *apn1Δ apn2Δ* are disrupted in combination with *csm2Δ* (Published Figure 4B on left).

For this paper, we tried to perform COMET assays but were unable to detect tails upon DSB formation (we tried very high IR doses).

Together, the strong *in vitro* data here combined with the *in vivo* data presented in this manuscript and in the literature is consistent with a novel function for the Shu complex in damage tolerance.

Reviewers' comments:

Reviewer #1 (Remarks to the Author):

NCOMMS-18-14750B

Rosenbaum et al.

The most important result in the paper is that "Csm2-Psy3" binds preferentially a double flapped DNA containing an apurinic/apirymidic (AP) site in vitro (Fig. 4 and Extended Fig. 5). In first and second rounds of reviewing, I asked the authors to explain the different affinity of the Csm2-Psy3 dimer to AP6 and AP7. In the other word, why the shows higher affinity to AP6 than to AP7? Since the both DNA substrates are structurally almost same (except the relative position of an AP site). Unfortunately, the authors could not get my point and as a result could not explain the discrepancy. So I describe the structure of AP6 and AP7 (and AP2 and AP3) in the attachment. In terms of DNA substrates containing an AP site, one expect that Csm2-Psy3 would show the same affinity to the both of AP6 and AP7 and higher than the substrate without the AP site (control) or AP2, AP3. However, Csm2-Pys3 shows the same affinity (Fig. 4) to AP7 and the control and other substrates, but does higher affinity to AP6 than AP7 and the others. This does NOT support the authors' claim that Csm2-Psy3 binds preferentially a double flapped DNA containing an AP site. More confusion is coming from new data (comments #2 and Extended data Fig. 5). EMSA shows the same affinity of Csm2-Psy3 to both AP6 and AP7 (the same concentration of the dimer shows similar amounts of band shifts to the two substrates, AP7 seems to be a bit better than AP6). This result is clearly different from the data in Fig. 4a. On the other hand, the result that AP6 and AP7 show similar affinity to the Csm2-Pys3 looks good in respect of their conclusion. However, unfortunately, the EMSA assay does not contain a control experiment with DNA without the AP site and/or AP2, 3 (as a result, one cannot conclude that Csm2-Pys3 bind preferentially to DNA with AP site).

Since data from these biochemical experiments are a key (novel claim) for this paper (in vivo data are just complementary), the authors should resolve this issue clearly.

Minor point:

As with authors' comment #1, I do not agree that ChIP is more sensitive than chromatin fractionation. As the authors noted, ChIP have several points the researchers pay more attention to. In the experiment of new data of ChIP in Extended data Fig. 10, how the authors repeat the experiments to get the standard errors (SE)? Are they biological replicates or technical replicates and how they calculate SE? What are statistical differences? Since an error bar in the graphs shows standard error (SE), it is useless to say statistical difference. The authors should show a confidential interval (CI; mean \pm 2SE) or other way (P-value). If one can use the confidential interval, it seems to me that ARS305 and ARS607, but not ARS216 and AR302, show significant binding to the protein relative to control. If this is true, the authors need more complicated explanation.

Reviewer #3 (Remarks to the Author):

Major problem in the current manuscript is lack of correlation with in vitro and in vivo data. Authors by in vitro assays show that Csm2/Psy3 prevents DSB formation at fork sites by AP endonuclease. This is not supported by in vivo data. They see an opposite phenotype. Authors should either remove Fig. 5 in vitro data or provide justifiable explanation in the discussion section.

Response to Reviewer Comments

Reviewer 1:

The most important result in the paper is that “Csm2-Psy3” binds preferentially a double flapped DNA containing an apurinic/apirymidic (AP) site in vitro (Fig. 4 and Extended Fig. 5). In first and second rounds of reviewing, I asked the authors to explain the different affinity of the Csm2-Psy3 dimer to AP6 and AP7. In the other word, why the shows higher affinity to AP6 than to AP7? Since the both DNA substrates are structurally almost same (except the relative position of an AP site).

Unfortunately, the authors could not get my point and as a result could not explain the discrepancy. So I describe the structure of AP6 and AP7 (and AP2 and AP3) in the attachment. In terms of DNA substrates containing an AP site, one expect that Csm2-Psy3 would show the same affinity to the both of AP6 and AP7 and higher than the substrate without the AP site (control) or AP2, AP3. However, Csm2-Pys3 shows the same affinity (Fig. 4) to AP7 and the control and other substrates, but does higher affinity to AP6 than AP7 and the others. This does NOT support the authors’ claim that Csm2-Psy3 binds preferentially a double flapped DNA containing an AP site.

More confusion is coming from new data (comments #2 and Extended data Fig. 5). EMSA shows the same affinity of Csm2-Psy3 to both AP6 and AP7 (the same concentration of the dimer shows similar amounts of band shifts to the two substrates, AP7 seems to be a bit better than AP6). This result is clearly different from the data in Fig. 4a. On the other hand, the result that AP6 and AP7 show similar affinity to the Csm2-Pys3 looks good in respect of their conclusion. However, unfortunately, the EMSA assay does not contain a control experiment with DNA without the AP site and/or AP2, 3 (as a result, one cannot conclude that Csm2-Pys3 bind preferentially to DNA with AP site).

Since data from these biochemical experiments are a key (novel claim) for this paper (in vivo data are just complementary), the authors should resolve this issue clearly.

We agree with Reviewer #1 that it is surprising that Csm2-Psy3 have different affinities for AP6 and AP7. These differences were reproducible between multiple protein preps and were done by two different individuals in different laboratories. Dr. Rosenbaum performed the initial experiments in Figure 4 in Dr. Andrew VanDemark’s laboratory. Each individual experiment contains three biological replicates but the experiment itself was performed three times using multiple protein preparations. Subsequently, Dr.

Hengel from my laboratory repeated these experiments using purified protein from our laboratory and that of Dr. Patrick Sung using the full Shu complex and again observed the same phenomenon of increased affinity of Csm2-Psy3-Shu1-Shu2 for AP6 vs WT.

It is possible that the different base pairing in the double-strand region adjacent to the AP site (AP6: G≡C vs AP7: A=T) can influence the structure of the fork or the conformation adopted by the Csm2-Psy3 heterodimer upon binding. Mechanistic understanding of the specificity differences between these two substrates would ideally be determined by atomic resolution structures, which is beyond the scope of the current manuscript. To date, published efforts have noted the inability to observe the Csm2-Psy3-DNA complex structure.

When we examined the nucleotide opposite the THF abasic site analog within AP6 and AP7 substrates, we performed EMSA experiments due to the number of unique oligonucleotides needed, the amount of protein required for anisotropy, and the time involved. The only conditions where we are able to visualize Csm2-Psy3 protein-DNA complexes within the acrylamide gel is in the presence of 5% glycerol and by running the gel at 4°C. Without glycerol addition, the protein remains in the wells. We respectfully disagree that Csm2-Psy3 bind with higher affinity to AP7 vs AP6. As observed and previously described [Garner, M.M. and Rau, D.C. (1995) *EMBO J.* 14:1257 and Vossen, K.M., et al. (1997) *Biochemistry* 36:1164], it is probable that the gel matrix and glycerol has stabilized the preformed complexes. This could potentially 1) slow the dissociation of the protein and DNA components or 2) maintain the Csm2-Psy3-DNA complex at levels as high or higher than achieved in equilibrium binding reactions. We have now extensively discussed these issues in the Results and Discussion and have softened the language throughout.

Please note that this biochemical experiment is supported by extensive *in vivo* and *in vitro* analysis.

1. **Figure 2b,c, Extended Data Figure 4b:** Csm2 chromatin association is enriched upon abasic site accumulation but not fork stalling.
2. **Figure 2a,b:** Csm2 DNA binding is required for its chromatin association when abasic sites accumulate and these mutants exhibit extreme DNA damage sensitivity and are highly mutagenic.
3. **Figure 3:** The *csm2Δ* and *psy3Δ* mutants exhibit mutation signatures consistent with abasic site repair on the lagging strand.
4. **Figure 5:** Csm2-Psy3 protect AP6 double-flap substrates from endonuclease cleavage.

Together, with the Csm2-Psy3 binding specificity characterization, we are extremely confident that the Shu complex has an important role in tolerance of abasic sites.

Minor Comment:

As with authors' comment #1, I do not agree that ChIP is more sensitive than

chromatin fractionation. As the authors noted, ChIP have several points the researchers pay more attention to. In the experiment of new data of ChIP in Extended data Fig. 10, how the authors repeat the experiments to get the standard errors (SE)? Are they biological replicates or technical replicates and how they calculate SE? What are statistical differences? Since an error bar in the graphs shows standard error (SE), it is useless to say statistical difference. The authors should show a confidential interval (CI; mean \pm 2SE) or other way (P-value). If one can use the confidential interval, it seems to me that ARS305 and ARS607, but not ARS216 and AR302, show significant binding to the protein relative to control. If this is true, the authors need more complicated explanation.

We have now removed the ChIP experiment from the manuscript.

Reviewer 3:

Major problem in the current manuscript is lack of correlation with in vitro and in vivo data. Authors by in vitro assays show that Csm2/Psy3 prevents DSB formation at fork sites by AP endonuclease. This is not supported by in vivo data. They see an opposite phenotype. Authors should either remove Fig. 5 in vitro data or provide justifiable explanation in the discussion section.

We agree with Reviewer 3 that if our model is correct, then we would expect to observe more DSBs upon MMS treatment when the Shu complex is disrupted. In budding yeast, comet assays are not typically performed likely because the genome is much smaller than human cells. Even though we perform comet assays routinely in our lab for mammalian work, we tried extensively to get this technique to work in budding yeast using high IR doses and we were unable to visualize any tail moment. As mentioned previously and shown in the prior review, we have already performed PFGE analysis and observed a delay in chromosome reconstitution that is consistent with more DSBs [Godin et al (2016) NAR 30(44): 8199]. In addition, as we previously mentioned, Shu complex disruption results in increased Rad52 foci, which are also indicative of increased DSBs [Shor et al (2005) Genetics 169(3): 1275]. Furthermore, we have also previously published using a direct repeat recombination assay that upon MMS treatment there are more recombination events and an increased in single-strand annealing likely due to more DSBs upon Shu complex disruption [Godin et al (2016) NAR 30(44): 8199]. We have now softened the language to indicate that our results provide *in vitro* evidence that the Shu complex protects against AP endonuclease cleavage.

However, we respectfully disagree with Reviewer 3 that our *in vivo* data showing synthetic sickness of *csm2Δ apn1Δ apn2Δ ntg1Δ ntg2Δ* contradicts the idea that the Shu complex protects from DSBs. We expect that the lethality observed in the Shu complex mutants combined with AP endonuclease and AP lyase mutants is likely due to the 1075X increase in spontaneous mutation rates that are observed [Godin et al (2016) NAR 30(44): 8199]. The mutational burden is extremely high and is likely resulting in cell lethality upon damage exposure. Consistent with this model, we find that the Shu complex mutants exhibit synthetic lethality when combined with TLS mutant *rev3Δ* (Pol zeta) [Godin et al (2016) NAR 30(44): 8199]. We have now discussed these finding in more depth in the Discussion.

In addition, it is important to note that new important findings from David Cortez's group demonstrate that a new protein HMCES, in human cells, binds abasic sites at a replication fork and protects against AP endonuclease cleavage by forming a crosslink [Mohni et al (2019) Cell 176: 144]. Our findings here are consistent with these results. In the case of the Shu complex, we find that the Shu complex mediates a template switching mechanism to avoid AP endonuclease cleavage to enable error-free lesion bypass. Together these results are consistent with our model, our past findings, and the current literature.